# High-frequency climate variability in the Holocene from a coastal-dome ice core in East-Central Greenland

Abigail G. Hughes[1], Tyler R. Jones[1], Bo M. Vinther[2], Vasileios Gkinis[2], C. Max Stevens[3], Valerie Morris[1], Bruce H. Vaughn[1], Christian Holme[2], Bradley R. Markle[4], and James W. C. White[1]

[1]Institute of Arctic and Alpine Research, University of Colorado Boulder, Boulder, Colorado, USA
[2]Physics of Ice, Climate and Earth, The Niels Bohr Institute, University of Copenhagen, Copenhagen, Denmark
[3]Department of Earth and Space Sciences, University of Washington, Seattle, Washington, USA
[4]California Institute of Technology, Pasadena, California, USA

**Correspondence:** Abigail G. Hughes (abigail.hughes@colorado.edu)

**Abstract.** An ice core drilled on the Renland Ice Cap in East-Central Greenland contains a continuous climate record dating through the last glacial period. The Renland record is valuable because the coastal environment is more likely to reflect regional sea surface conditions, compared to inland Greenland ice cores that capture synoptic variability. Here we present the $\delta^{18}$O water isotope record for the Holocene, in which decadal-scale climate information is retained for the last 8 ka, while the annual water isotope signal is preserved throughout the last 2.6 ka. To investigate regional climate information preserved in the water isotope record, we apply spectral analysis techniques to a 300-year moving window to determine the mean strength of varying frequency bands through time. We find that the strength of 15–20 year $\delta^{18}$O variability exhibits a millennial-scale signal in line with the well-known Bond events. Comparison to other North Atlantic proxy records suggests that the 15–20 year variability may reflect fluctuating sea surface conditions throughout the Holocene, driven by changes in the strength of the Atlantic Meridional Overturning Circulation. Additional analysis of the seasonal signal over the last 2.6 ka reveals that the winter $\delta^{18}$O signal has experienced a decreasing trend, while the summer signal has predominantly remained stable. The winter trend may correspond to an increase in Arctic sea ice cover which is driven by a decrease in total annual insolation, and is also likely influenced by regional climate variables such as atmospheric and oceanic circulation. In the context of anthropogenic climate change, the winter trend may have important implications for feedback processes as sea ice retreats in the Arctic.

## 1 Introduction

Ice core records are powerful archives of past climate change, containing hundreds of thousands of years of climate information. Greenland ice core records are valuable for determining a comprehensive picture of regional North Atlantic climate patterns throughout the Holocene and last glacial period. Recent developments in Continuous Flow Analysis and Cavity Ring-Down Spectroscopy allow for detection of high-frequency signals in ice core water isotopes. These measurement techniques were used in analysis of the REnland ice CAP project (RECAP) ice core, located on a coastal dome in East-Central Greenland. The RECAP ice core extends 584 m to bedrock, with the oldest ice dating 120 ka; here we present the Holocene water isotope record (i.e. $\delta^{18}$O) (Fig. 2). Polar ice core water isotope records are correlated to condensation temperature at the time of precipitation

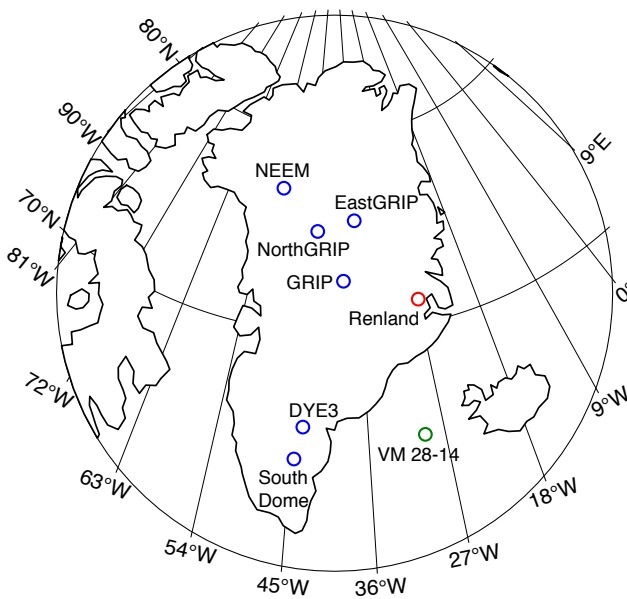

**Figure 1.** A map of the study region shows the RECAP drill site (red) is located on the east coast of Greenland. The Renland ice cap is approximately 80 km wide, and is isolated from the Greenland ice sheet. The drill site is near the summit of the ice cap, at a location of -26.75, 71.2333. The locations of several other Greenland ice cores are also shown for reference (blue), as well as North Atlantic sediment core VM 28-14 (green) (Pawlowicz, 2020).

(Dansgaard, 1964; Dansgaard et al., 1973; Craig and Gordon, 1965; Merlivat and Jouzel, 1979; Jouzel and Merlivat, 1984; Jouzel et al., 1997), and integrate across regional ocean and atmospheric circulation patterns and sea surface conditions along
the moisture transport pathway (Johnsen et al., 2001; Holme et al., 2019).

The Renland Peninsula is located on the eastern coast of Greenland in the Scoresbysund Fjord (Fig. 1). The ice cap is unique in that it is isolated from the Greenland Ice Sheet by steep fjords and is only 80 km wide; as a result, the thickness of the ice cap is constrained and did not experience significant change during most of the Holocene (Vinther et al., 2009; Johnsen et al., 1992). The Renland Peninsula experienced post-glacial uplift in the early Holocene due to ice sheet retreat, but the rate of uplift
has been minimal over the last 7 ka (Vinther et al., 2009). These factors imply that the climate record is not influenced by long-term changes in elevation through the mid to late Holocene. The modern accumulation rate at Renland is approximately 45 cm ice equivalent accumulation per year, resulting in clearly defined annual layers. In comparison to inland ice core records, the local climate at the coastal Renland site is more likely linked to sea surface conditions and North Atlantic climatology (Holme et al., 2019; Johnsen et al., 2001).

The modern instrumental record documents a number of influences on climate variability in the North Atlantic and Arctic, which may exert an influence on the local climate at Renland. The Atlantic Meridional Overturning Circulation (AMOC) controls heat transport to the Arctic and can have a substantial effect on Arctic climate over long timescales. Heat is supplied to the Arctic via the Norwegian Current, carrying warm Atlantic water to the Arctic (Polyakov et al., 2004); changes in heat supply

and northward advection will influence atmosphere-ocean heat exchange, regional Arctic climate, and sea ice cover (Muilwijk et al., 2018). Heat distribution through AMOC controls sea surface temperature and drives the Atlantic Multidecadal Oscillation (AMO), which is observed in prior Greenland ice cores and influences sea ice cover in the Arctic (Chylek et al., 2011) with approximately 20-year variability. Subdecadal climate signals are also observed in the Arctic, such as the North Atlantic Oscillation (NAO), which is currently expressed as shifting sea-level pressure differences between the Subtropical High and Subpolar Low. This multi-year variability influences temperature, precipitation, sea ice distribution, and ocean circulation across the entire North Atlantic (Hurrell and Deser, 2009), and it is recorded in multiple proxy records including tree-ring data and central and western Greenland ice cores (Barlow et al., 1993; Appenzeller et al., 1998).

On longer timescales, changes in sea surface conditions are also reflected in North Atlantic sediment cores. Sediment cores from the North Atlantic show that millennial-scale climate variability occurred throughout the Holocene (Bond et al., 1997, 2001), referred to as Bond events. The mechanism forcing this variability is still under debate, but it is potentially driven by solar forcing (Bond et al., 2001) or internal climate dynamics such as interactions between the ocean and atmosphere (Wanner et al., 2015). The effects are most prominent in the Arctic, likely transmitted to lower latitudes through AMOC (Bond et al., 1997, 2001; DeMonocal et al., 2000).

By reconstructing a comprehensive climate record of the Holocene, we can better place modern anthropogenic climate change in context of the past. Using the $\delta^{18}$O record from the RECAP ice core, we aim to determine the factors that influence regional North Atlantic climate evolution over the Holocene. We explore interannual- to decadal-scale oscillations in $\delta^{18}$O over the last 8 ka, and investigate relationships to sediment core records in the North Atlantic. We also calculate changes in $\delta^{18}$O seasonality throughout the last 2.6 ka, the time period for which the annual signal can be resolved, in order to determine driving factors for summer and winter temperatures. This analysis will give insight to mechanisms which influence regional climate in coastal Greenland over several timescales.

## 2    Methods

### 2.1    Ice core analysis

The RECAP ice core was drilled near the summit of the Renland ice cap (-26.75, 71.2333) from May–June 2015. The drilling location is approximately 2 km away from the location of a core previously drilled in 1988 (Johnsen et al., 1992). The 584 m RECAP core, drilled to bedrock, contains a continuous record extending through 120 ka. The water isotope record was measured using a Continuous Flow Analysis (CFA) system (Gkinis et al., 2011; Jones et al., 2017b) at the Center for Ice and Climate at the University of Copenhagen. A stick of ice is continuously melted at a rate of 2–4 cm min$^{-1}$, and meltwater is collected, vaporized, and analyzed using two Cavity Ring-Down Spectrometers running in parallel (Picarro L2140-$i$ and L2130-$i$). This technique produces $\delta^{18}$O, $\delta$D, and $\delta^{17}$O water isotopes, where delta notation refers to a ratio of heavy to light isotopes measured with respect to a standard (Vienna Standard Mean Ocean Water) and is expressed in parts per thousand (per mille or ‰) (Dansgaard, 1964). Water isotope data has sub-mm nominal resolution, as there is a small amount of mixing introduced in the system due to liquid water mixing in tubing or vapor mixing (Gkinis et al., 2011; Jones et al., 2017a). High-

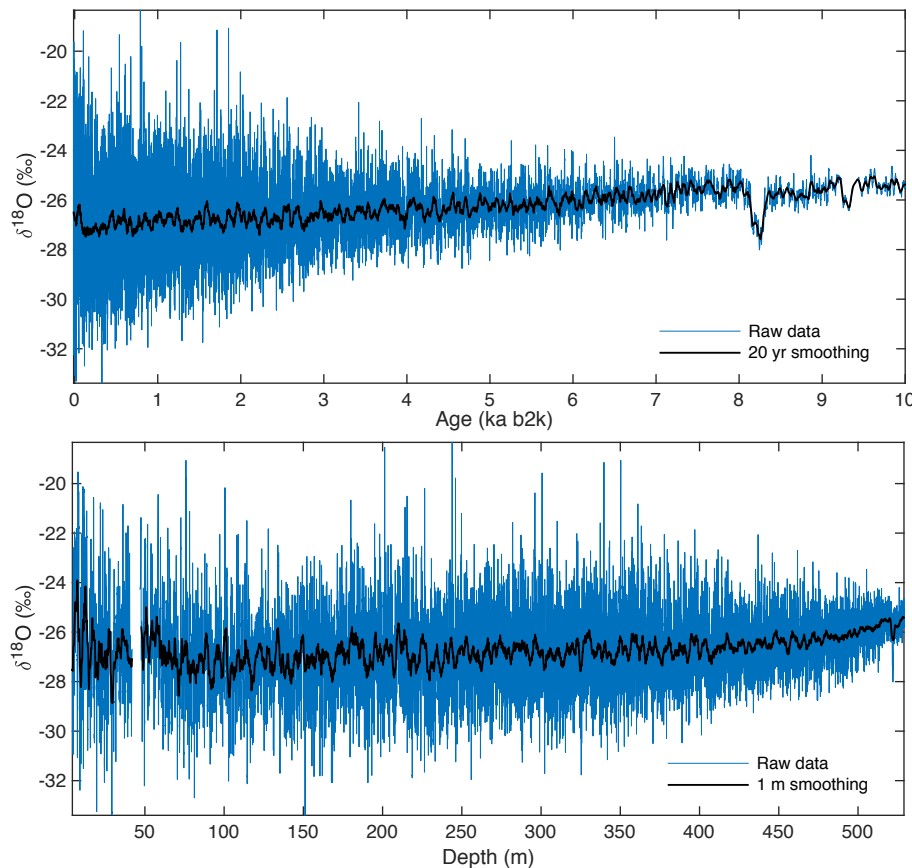

**Figure 2.** (Top) Holocene RECAP $\delta^{18}O$ data in time (raw, blue; 20 yr smoothing, black); and (Bottom) RECAP $\delta^{18}O$ data in depth (raw, blue; 1 m smoothing, black). The loss of high-frequency data is visually observed in the shape of the $\delta^{18}O$ record, which tapers rapidly as the signal is diffused. At a depth of 529 m, the age of the ice is 10 ka.

frequency data is then bin-averaged into 0.5 cm data points. We focus on the $\delta^{18}O$ record in this paper. Small gaps in data, inherent to the CFA methodology due to breaks in ice and transitions from standards (Jones et al., 2017b), are infilled using a maximum entropy method spectral technique (Andersen, 1974; Fahlman and Ulrych, 1982).

The depth-age scale for the RECAP ice core is determined by Simonsen et al. (2019). For the period from -13–4048 yr b2k (years before CE 2000), the StratiCounter algorithm of annual layer counting was applied (Winstrup et al., 2012; Winstrup, 2016). For ages greater than 4048 yr b2k, a shape-preserving piecewise cubic interpolation is used based on Greenland Ice Core Chronology 2005 (GICC05) reference tie points (Simonsen et al., 2019). The full time scale is also fit to a series of tie points using the GICC05 reference timescale (Simonsen et al., 2019).

The RECAP record exhibits a much higher effective resolution in the Holocene than during the glacial period. High-frequency variability is lost rapidly with depth due to an increase in the effect of water isotope diffusion, which occurs for two reasons: 1) At the last glacial maximum, the Laurentide ice sheet extended to approximately 40° N, and the associated

temperature decrease and sea ice increase led to significantly drier conditions with minimal precipitation in Greenland. As a result, water isotope diffusion has a much greater effect in thinner glacial ice layers, acting to eliminate high-frequency signals.

2) Extreme basal thinning (see Fig. A1) effectively deforms annual layers into very thin intervals of ice, allowing solid-phase water isotope diffusion to have a disproportionately strong effect on the oldest ice. The glacial period is condensed into approximately 30 m, and no high-frequency climate signals (i.e. annual, interannual, decadal) are preserved. As a result, we focus here on the Holocene record through 10 ka, which has retained greater high-frequency climate information.

## 2.2    Vapor diffusion

In the firn column, vapor diffuses along concentration and temperature gradients. Exchange of water molecules takes place between unconsolidated snow grains and vapor, attenuating the seasonal isotopic signal and acting as a smoothing function (Whillans and Grootes, 1985; Cuffey and Steig, 1998; Johnsen et al., 2000; Jones et al., 2017a). Solid-phase water isotope diffusion in ice below the firn column occurs at a much slower rate, with diffusivities increasing for warmer ice near bedrock. Over thousands of years, solid-phase diffusion can have a substantial impact on the attenuation and smoothing of high-frequency
signals (Itagaki, 1967; Robin, 1983; Johnsen et al., 2000; Gkinis et al., 2014; Jones et al., 2017a).

At the pore close-off density of $\sim$804 kg m$^{-3}$ (Johnsen et al., 2000; Jean-Baptiste et al., 1998), the transition between firn and ice occurs. The age of the ice at which this occurs is influenced by accumulation rate and temperature, and at Renland is 63 yr b2k (56 m below surface). All following calculations do not include the period -13–63 yr b2k, because the extent of diffusion varies substantially in this interval.

The extent of diffusion is characterized by the diffusion length parameter $\sigma_z$ (units meters), which represents the mean displacement of water molecules from their relative original position (Johnsen et al., 2000). Spectral analysis is used to estimate the diffusion length, including effects of firn diffusion, solid-phase diffusion, and CFA system mixing. The Multi Taper Method (MTM) of Fourier transform is applied to sections of the water isotope record in the depth domain (i.e. 0.5 cm resolution), using overlapping windows corresponding to 300-year time periods with a step size of 100 years between windows. This produces
the power spectral density (PSD) ($‰^2$m) vs. frequency ($f$) (m$^{-1}$) for each 300-year window. Vapor diffusion in the firn column causes the PSD to progressively decrease at higher frequencies, taking the form of quasi-red noise. To estimate the diffusion length, we apply the following methods described in Jones et al. (2017a), similar to Kahle et al. (2018) and Holme et al. (2019). The PSD of the ice core data can be described by an exponential decay model assuming Gaussian measurement noise:

$$P(f) = P_0(f) \cdot \exp(-(2\pi f \sigma_z)^2) + N(f) \tag{1}$$

where $P(f)$ is the diffused PSD derived from the raw water isotope data, $P_0(f)$ is the signal prior to diffusion, and $N(f)$ is measurement noise. Eq. 1 is fit to the diffused portion of the PSD to estimate $\sigma_z$ (Fig. 3a, diffused section indicated by blue line).

To estimate the uncertainty range for $\sigma_z$, a linear regression is calculated for $\ln(PD)$ vs $f^2$ of the diffused section of data (Fig. 3b). We find the maximum and minimum slope ($m_{lr}$) within one standard deviation of the linear fit:

$$\hat{\sigma_z} = \frac{1}{2\pi\sqrt{2}} \cdot \left(\frac{1}{2|m_{lr}|}\right)^{-\frac{1}{2}} \tag{2}$$

where $\hat{\sigma}_z$ represents one standard deviation of the diffusion length.

Improvements to the fitting routine have been made to account for a higher density of data points at high frequency, which biases the fit of the Gaussian to high frequencies. The data is binned and averaged evenly in log-space frequency, and the Gaussian is fit to averaged data points using a least-squares optimization. Finally, the diffusion length in meters is converted to time using the following equation:

$$\sigma_t = \frac{\sigma_z}{\lambda_{avg}} \tag{3}$$

where $\lambda_{avg}$ is the mean annual layer thickness (m yr$^{-1}$) for the given window. The diffusion length in the depth domain (Fig. 3c) rapidly decreases with the age of ice, because the deeper layers of the Renland ice cap are subject to extreme thinning. As ice layers are buried and compacted, water molecules which were dispersed in the firn via vapor diffusion are brought closer towards their original relative position in the ice due to thinning. However, the diffusion length in the time domain (Fig. 3d) increases with age. Because the effects of firn diffusion are locked in after the pore close-off depth, an increase in diffusion length with time could be potentially due to effects of solid-phase diffusion. Alternatively, changes in surface conditions (i.e. temperature, accumulation rate, air pressure) could have influenced the extent of firn diffusion at the time of deposition (Whillans and Grootes, 1985; Johnsen et al., 2000).

Reliable diffusion length estimation can be applied to $\delta^{18}O$ data over the range $\sim$0.1–8 ka. However, there are two problematic sections from 5.6–6.7 and 7.8–8 ka, in which the power spectral model fails to accurately describe the power spectral densities, adding uncertainty to the diffusion length estimation. We compare two fitting routines for all spectra from 5.6–8 ka, hereafter referred to as Fit 1 and Fit 2 (see Fig. A3 for examples and further explanation of fits). As demonstrated in Fig. 3, the different fitting scenarios produce a minimal change in the diffusion length estimate. Even so, both fitting scenarios are used in all diffusion-correction calculations in this paper, in order to ensure that results are not influenced by bias in the diffusion length estimate.

## 2.3 Signal decay of individual frequencies

Further analysis of the PSD can be used to determine the amplitude of climate signals on an interannual to decadal scale (Jones et al., 2018). While the raw isotope data is comprised of a continuum of climate variability at all frequencies, it is possible to separate out the strength (ie. the amplitude) of varying frequency bands within that continuum. The power ($P_i$) of individual frequencies from 1–20 years are identified for each 300-year PSD window, as well as the frequency bands of 3–7, 7–15, 15–20, and 20–30 years. For frequency bands, the average power is calculated by integrating across the power spectrum within each

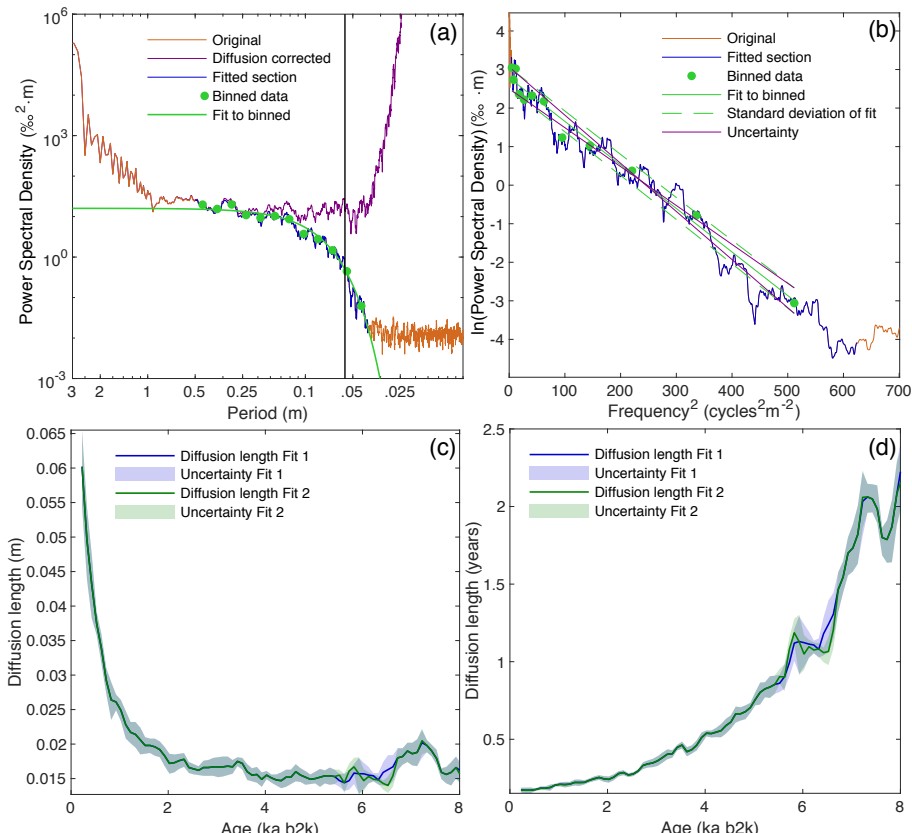

**Figure 3.** Isotope power spectra and diffusion length estimation. (a) The power spectral density for a 300-year window from 2263–2563 yr b2k. The diffused section of the spectrum is fit with a Gaussian (Eq. 1), used to estimate diffusion length for each window. The signal is cut off at a frequency of 1.1 yr$^{-1}$ (indicated by vertical black line), after which point it is primarily noise. At periodicities below the cut-off frequency, diffusion correction of noise results in an unstable signal (as shown by the purple PSD to the right of the cut-off frequency). (b) Natural log of the same PSD, used to estimate uncertainty. (c) Diffusion length estimated in depth and (d) converted to time. In both diffusion length figures, the two fitting scenarios and their respective uncertainty is designated by blue (Fit 1) and green (Fit 2). The section from 5.6–6.7 ka has the lowest quality spectra, resulting in a small difference in the diffusion length estimates.

frequency band:

$$P_i = \frac{\int_{f_a}^{f_b} P(f)\,\mathrm{d}f}{f_b - f_a} \tag{4}$$

where $f_a$ and $f_b$ represent the lower and upper frequency limits, respectively. Because the frequencies are not evenly spaced, this method ensures that the average is not biased towards higher frequencies. The amplitude (ie. the strength) is calculated as the square root of $P_i$. In our analysis, the strengths of individual frequencies are normalized to the strength of the annual signal

in the most recent window, and the strength of each frequency band is normalized to its most recent value. This produces a time series of the relative strength of isotopic variability for several individual frequencies (1, 3, 5, 7, 10, 20 years) and interannual to decadal frequency bands (3–7, 7–15, 15–20, and 20–30 years) (Figs. 5, 6).

The pre-diffusion strength of each frequency band is estimated from Eq. 1, with the correction carried out using diffusion lengths calculated by both Fit 1 and Fit 2. Due to uncertainty in the diffusion length estimation, the 3–7 and 7–15 year bands can be corrected for diffusion over the range ∼0.1–5.5 ka; over this time period uncertainty is better constrained. The decadal bands (15–20 and 20–30 years) are substantially less affected by diffusion and can be corrected from ∼0.1–8 ka.

## 2.4   Deconvolution and identification of seasonal signal

Deconvolution is used to correct for the effects of diffusion on the original water isotope signal as it existed at the surface of the ice sheet, including effects of firn diffusion, solid-phase diffusion, and CFA system mixing (Johnsen et al., 2000; Vinther et al., 2003). An estimate of the original power spectrum (i.e. $P_0'$, where $P_0$ is the original power spectrum prior to diffusion, and $P_0'$ is our estimate of it) is obtained by using the diffusion length estimates for each 300-year window, producing a diffusion-corrected power spectrum with noise removed below a cut-off frequency of 1.1 yr$^{-1}$ (as demonstrated in Fig. 3a); therefore, we can assume that the term $N(f)$ in Eq. 1 is insignificant for the frequencies below the cut-off and is equal to zero. The corrected noise-free power spectral density is then inverted to the time domain to approximate the deconvolved $\delta^{18}$O record for each window (Vinther et al., 2003). A comparison between the raw data and the diffusion-corrected signal from 63–2600 yr b2k is shown in Fig. A2. The diffusion correction is also carried out using diffusion length uncertainty values (Fig. 4).

A peak detection algorithm (built-in Matlab function findpeaks) is used to select extrema (summers and winters) in the diffusion-corrected $\delta^{18}$O signal from 63–2600 yr b2k (Fig. 4). A 40-year moving average is applied to the resulting summer and winter time series to filter out high-frequency noise and distinguish long-term trends (Fig. 8).

## 2.5   Community Firn Model

Climate variability such as changes in temperature and accumulation rate can influence the extent of diffusion that occurs in the firn column. While the method of estimating diffusion length directly from the water isotope record includes the effects of long-term changes in mean temperature and accumulation rate, changes in seasonality of accumulation creates uncertainty in the diffusion-correction calculation of the annual cycle. If there is a seasonal accumulation bias (i.e. more snow in summer than winter), the drier season will be subject to greater isotopic attenuation due to firn diffusion. Because we cannot selectively diffusion-correct the seasonal isotope signal for accumulation bias, we must assume constant seasonality of accumulation. Therefore, the drier season will be under-corrected for diffusion, and the wetter season will be over-corrected to a lesser extent, resulting in potential inaccuracies in the amplitude of the seasonal signal. While there is no current method for reconstructing past seasonality of accumulation, we can utilize a series of tests to determine the extent to which the diffusion-correction calculation for $\delta^{18}$O could be affected by seasonally-biased accumulation.

We use the Community Firn Model (CFM) (Stevens et al., 2020) to test the effects of seasonally-biased accumulation on firn diffusion in the ice core. We test five 490-year scenarios for isotope evolution, based upon temperature and accumulation fields

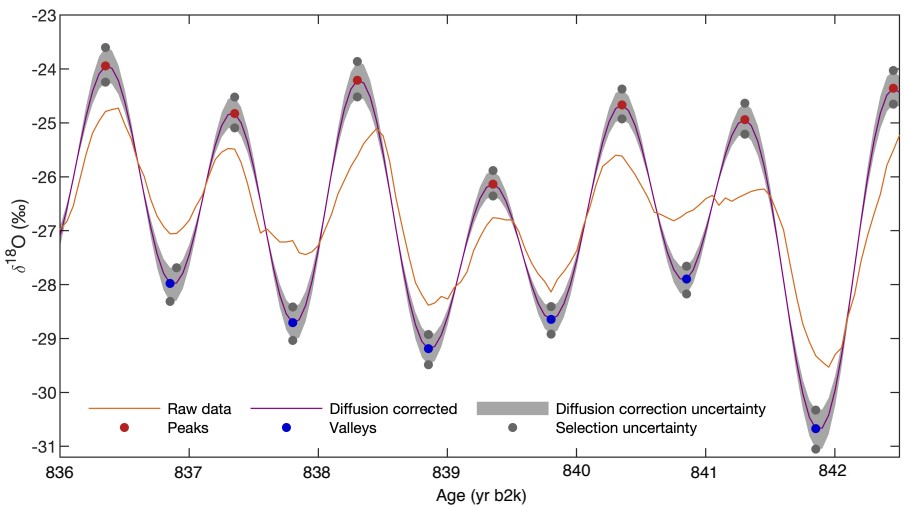

**Figure 4.** An example of diffusion-corrected data (purple) compared to raw data (orange), with summer maximum (red) and winter minimum (blue) selected for each annual signal. Gray shading and points indicate uncertainty.

from 1958–1978 provided by the Modèle Atmosphérique Régional (MAR; version 3.9 with monthly ERA forcing) (Fettweis et al., 2017) (Fig. A4a–b). A constant amplitude (4‰) sine wave is used to represent the annual isotopic variability (Fig. A4c), based on the mean amplitude of the relatively un-diffused most recent 10 years of the $\delta^{18}O$ signal. The mean monthly 1958–1978 temperature cycle (Fig. A4e) from MAR is used for all model scenarios. For the 20-year period, MAR simulates that the summer months July–September receive the most accumulation on average (Fig. A4d). The mean annual accumulation from the 20-year MAR period is used for all model scenarios, with the following variations on the seasonality of accumulation applied: (1) 'Constant': Constant values for monthly accumulation rate (i.e. each month receives the same amount of accumulation); (2) 'Cycle': Repeating annual cycle, with each month's accumulation value assigned the 20-year mean accumulation rate from MAR for that month (Fig. A4d); (3) 'Noise': As in (2), but with noise added to the cycled values; the statistics of the noise are derived from the standard deviation of the 20-year MAR time series for each month; (4) 'Random': Random value selected from the normal distribution of each month's 20-year MAR time series; and (5) 'Loop': Repeating (looped) 20-year intervals of MAR data.

   The temperature, accumulation, and synthetic isotope data are input to the CFM to produce an estimate of how the isotopic signal is diffused within the firn column over a 490-year period. The resulting isotope record represents the effects of firn densification (Kuipers Munneke et al., 2015), vapor diffusion (Johnsen et al., 2000), and climate conditions on an ice core water isotope record. In our analysis, the diffused isotope signal from 390–490 years is selected, because this is below the bubble close off depth of the firn column. As with the observed ice core data, we diffusion-correct the model data, and select the summer and winter extrema values.

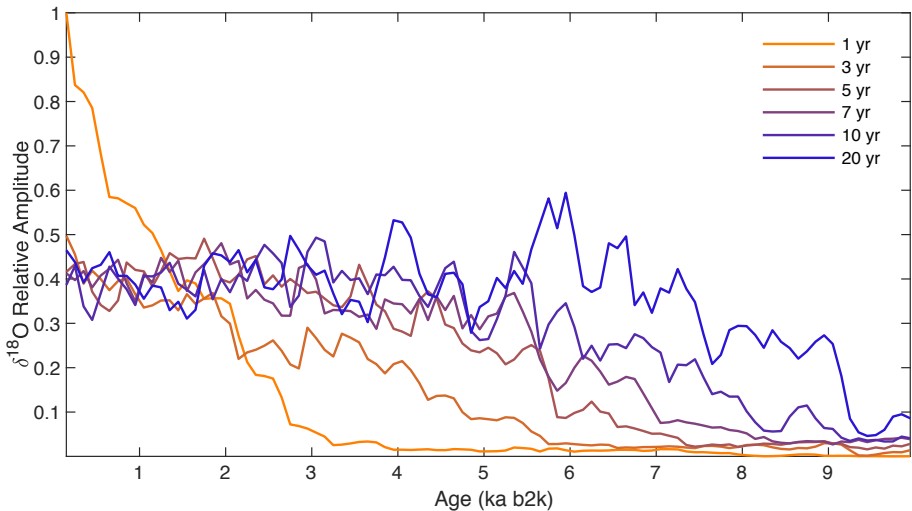

**Figure 5.** The relative amplitude of selected individual frequency bands from 1–20 years is shown for the RECAP $\delta^{18}$O signal, with all bands normalized to the strength of the annual frequency in the most recent window. This demonstrates how rapidly the strength of the annual signal decreases, while lower frequency signals are preserved for a greater period of time.

## 3 Results and discussion

In the following section, we present the results together with the discussion, separated to first look at the interannual climate variability (Section 3.1) and then late Holocene seasonality (Section 3.2). The analysis of interannual climate variability includes the relative amplitudes of individual 1–20 year frequencies and high-frequency bands, followed by a discussion considering the similarities observed between the 15–20 year relative amplitude band and North Atlantic sediment core VM 28-14. In Section 3.2, we first present the results of diffusion-correction and peak selection through 2.6 ka, identifying trends in seasonality. This is followed by analysis of several potential driving mechanisms, culminating in a discussion of regional climate variables.

### 3.1 Interannual climate variability

The strength of the annual signal in $\delta^{18}$O decays rapidly with time due to diffusion (Fig. 5), but persists to approximately 2.6 ka. After this time, the annual signal cannot be diffusion-corrected or interpreted. The strength of interannual signals also decreases with time; for example, the 3-year signal is lost as it decays below a relative amplitude of 0.05‰ at ~5.6 ka and the 5-year signal is lost at ~7 ka. The decadal signal is preserved for most of the Holocene, extending to about 8 ka (Fig. 5). Further analysis of frequency bands (3–7, 7–15, 15–20, 20–30 year bands) is also used to investigate variability of interannual and decadal climate signals (Fig. 6). The higher frequency bands are more diffused, and we find that the 3–7 and 7–15 year bands are not reliably corrected after 5.5 ka due to uncertainty in the diffusion length estimate (see Fig. 3). Fig. 6 demonstrates that the 15–20 and 20–30 year bands are less influenced by diffusion, as there is a minimal difference between the strength of

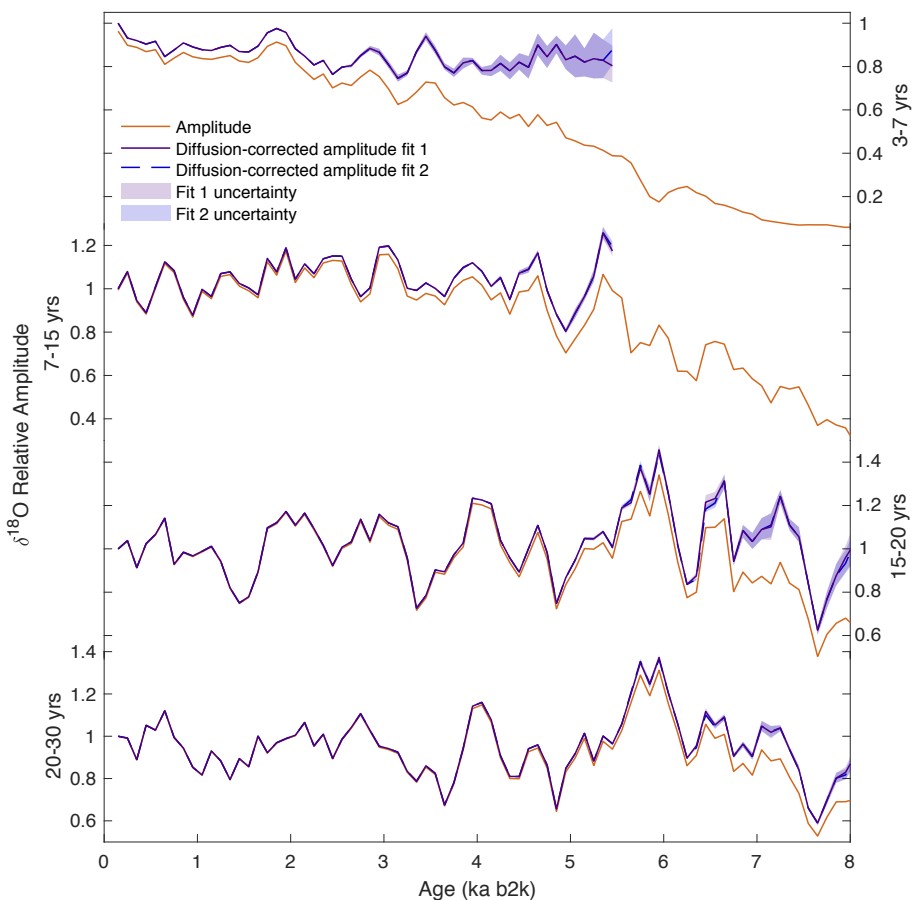

**Figure 6.** The normalized relative amplitude for 3–7, 7–15, 15–20, and 20–30 year bands (orange) decreases with time due to diffusion and thinning in the ice core. The solid and dashed purple lines indicate the diffusion-corrected relative amplitude, using two different fitting scenarios for the diffusion length estimate. The two fitting scenarios result in nearly identical diffusion-corrected signals, therefore much of the lines are overlapping. The strength of each band is normalized to the amplitude of the respective diffusion-corrected signal in the most recent window. The diffusion-correction of frequency bands 3–7 and 7–15 years is cut off at 5.5 ka due to uncertainties in the diffusion length. Decadal bands from 15–20 and 20–30 years are not as affected by diffusion, and are corrected through 8 ka. The point at which the diffusion-correction deviates from the original value shows that as the frequency decreases, the signal resists attenuation for a greater period of time.

the raw and diffusion-corrected data, and the diffusion-correction does not significantly influence the shape of the curve over 8 ka.

After an investigation of how frequency bands compare to other climate records, we find the 15–20 year band has distinct similarities to a record of hematite-stained grains (HSG) in North Atlantic sediment core VM 28-14 (Bond et al., 2001) (Fig. 1, 220 Fig. 7). The full span of the two records has a correlation coefficient $r = 0.34$, while the period from 2.5–6.4 ka has a stronger relationship with $r = 0.74$. Both relationships have a p-value $<0.01$, indicating that they are statistically significant, and the

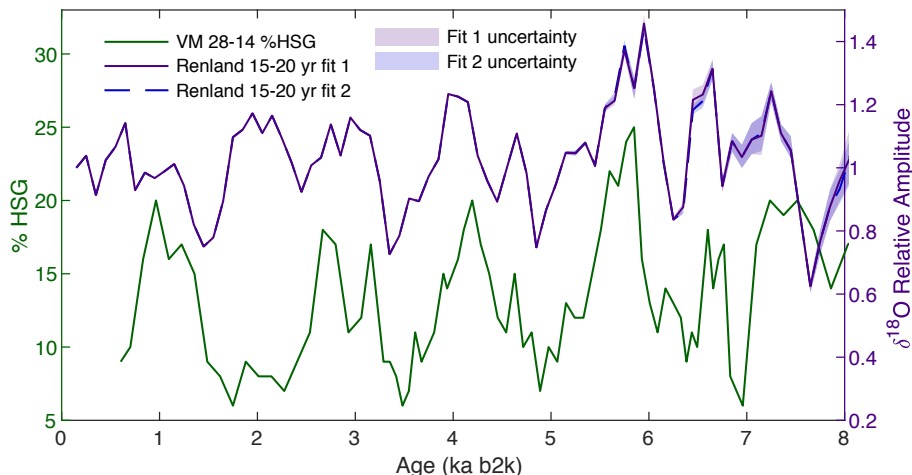

**Figure 7.** Comparison of %HSG from North Atlantic sediment core VM 28-14, a tracer for drift ice (Bond et al., 2001) and diffusion-corrected 15–20 year relative amplitude of $\delta^{18}O$ in the RECAP ice core, normalized to its starting amplitude. The period from 2.5–6.4 ka exhibits the strongest relationship, with correlation coefficient r = 0.74.

time series are not significantly autocorrelated. Additionally, it is possible that inaccuracies in the dating model for VM28-14 (on the order of $\pm$ 200–500 years) (Bond et al., 2001) could account for some of the reduced strength in correlation outside of the period from 2.5–6.4 ka. While we cannot rule out the possibility that this similarity arises due to random noise in the
225 climate system, considering the common geographic region, we can hypothesize how potential mechanisms could link the two records.

Percent HSG in an ocean sediment core is sensitive to the amount and source of glacial ice and sea ice, reflecting circulation in surface ocean waters. A decrease in HSG in core VM 28-14 is indicative of a greater influx of cold surface waters carrying drift ice from the Nordic seas, driven by strong northerly winds (Bond et al., 2001; Andrews et al., 2014). Millennial variability
observed in VM 28-14 is known as Bond events, but the mechanism driving these events has long been a source of debate. A comparison with Beryllium records in Greenland ice cores and carbon-14 in tree rings shows that increases in drift ice and HSG correspond to intervals of variable and decreased insolation (Bond et al., 2001), suggesting that changes in millennial-scale circulation are driven by insolation. However, more recent studies suggest that Bond events are unrelated to insolation and are due to internal climate system variability or volcanic activity (Wanner et al., 2015). While the driving mechanism remains
uncertain, it is known that percent HSG fluctuates with influxes of Nordic surface waters which flow along the eastern coast of Greenland and near the RECAP core location (Bond et al., 2001; Andrews et al., 2014).

Modern observations of North Atlantic variability may provide some clues as to why the amplitude of the 15–20 year band has apparent similarities with regional ocean circulation. One potential climate mechanism that could be expressed in the 15–20 year $\delta^{18}O$ variability is the North Atlantic Oscillation (NAO), as fluctuations in insolation could influence the strength
of the NAO through changes in sea surface temperature (Shindell et al., 2001). The NAO is observed in water isotope and

accumulation records of central and west Greenland ice core records through the 19th century, but the $\delta^{18}$O–NAO relationship on the east coast of Greenland is weak (Appenzeller et al., 1998; Barlow et al., 1993; Vinther et al., 2003, 2010).

Similarly, a recent analysis of the last 200 years of stacked ice core records from Renland demonstrated that $\delta^{18}$O shows varying levels of weak correlation with NAO (Holme et al., 2019). Instead, the $\delta^{18}$O signal is dominated by a combination of climate conditions including regional temperature and sea ice extent in the Fram Strait. During periods of increased sea ice extent, the $\delta^{18}$O–temperature relationship at Renland is weaker (Holme et al., 2019). Similarly, the 15–20 year variability throughout the Holocene may be responding to regional temperature, sea ice extent, and how these factors are influenced by interaction with ocean circulation.

The connection between sea surface conditions and ice core records is further supported by a previous study linking water isotope variability in a stack of five Greenland ice cores to the Atlantic Multidecadal Oscillation (AMO) using instrumental and proxy records (Chylek et al., 2011). Over the period from 1303–1961, spectral analysis shows 20-year variability in the ice core stack, also observed in prior model simulations of AMO (Chylek et al., 2011; Knight et al., 2005) (see Appendix Fig. A5 for additional spectral analysis similar to methods used in Chylek et al. (2011)). The correlation between prominent multidecadal variability in both Greenland ice core records and climate model simulations is attributed to changes in sea surface temperature (SST), driven by AMO and associated with variability in Atlantic Meridional Overturning Circulation (AMOC) in the North Atlantic basin (Chylek et al., 2011; Frankcombe et al., 2010). The SST and Arctic climate are sensitive to the strength of AMOC, as a reduction in overturning circulation would lead to decreased northward heat transport. These findings suggest that the variability in the 15–20 year $\delta^{18}$O signal at Renland may be associated with the AMO, driven by AMOC and the associated sea surface conditions including both ice cover and SST. Additionally, Knight et al. (2005) shows that the AMO signal observed in the HadCM3 model is non-stationary on millennial time scales, potentially explaining the variability in the strength of the 15–20 year signal at Renland. While the correlation between the marine HSG record and the RECAP ice core $\delta^{18}$O record is not conclusive evidence of the relationship between the Renland 15–20 year $\delta^{18}$O signal and AMO, there is a plausible physical connection between the Bond events and the decadal variability observed in Greenland. Additional studies that reduce timescale uncertainties in ocean sediment records, in combination with further modeling of the physical processes, would help to constrain the mechanisms linking the two records. Modeling in particular can elucidate the millennial-scale relationships between the East Greenland ice sheet, North Atlantic ocean dynamics, and regional climate patterns.

## 3.2  Late-Holocene seasonality

As part of the annual cycle, maximum summer and minimum winter $\delta^{18}$O values were determined using a selection algorithm for the period from 63–2600 years b2k (Fig. 8). The winter $\delta^{18}$O signal decreases toward present with the average modern value approximately 1.2‰ lower than the average value at 2.6 ka b2k. The summer signal trend over this same time period is relatively flat. Both summer and winter records also exhibit some variability on centennial timescales over the last 2.6 ka. The amplitude of the annual signal (half the summer to winter difference) has increased by approximately 50% from 2.6 ka to the present, indicating stronger seasonality in the modern Arctic climate system. Here we investigate several potential mechanisms that could drive a decrease in winter $\delta^{18}$O at Renland, and the associated amplitude increase.

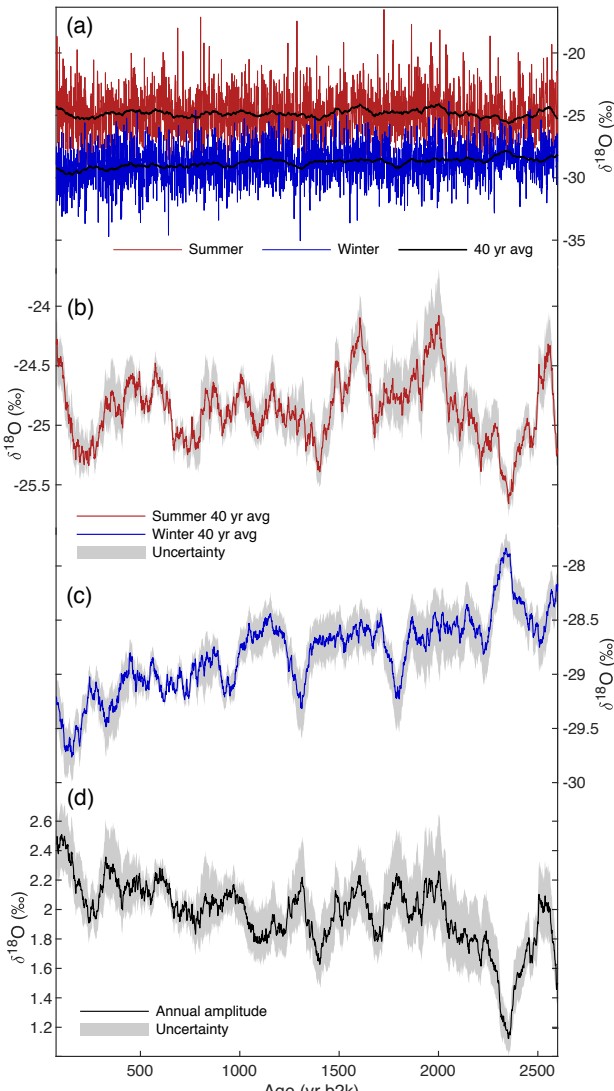

**Figure 8.** RECAP seasonal $\delta^{18}O$ data: (a) seasonal signal for diffusion-corrected data (summer is red; winter is blue) with a 40-year moving average applied in black; (b,c) 40-year moving average of seasonal signal with maximum and minimum uncertainty indicated by gray shading; (d) annual amplitude of seasonal signal with maximum and minimum uncertainty indicated by gray shading.

### 3.2.1 Seasonality of accumulation

We compare the diffusion-corrected outputs from the Community Firn Model (CFM) when it is forced with the five accumulation scenarios (constant, cycle, noise, random, loop) described in Section 2.5. The model results show that the RECAP diffusion-correction is minimally influenced by seasonality of accumulation (Fig. 9). This outcome is unique to the Renland site, which has a much higher accumulation rate (∼1.5 meters of snowfall per year) in comparison to other inland Greenland

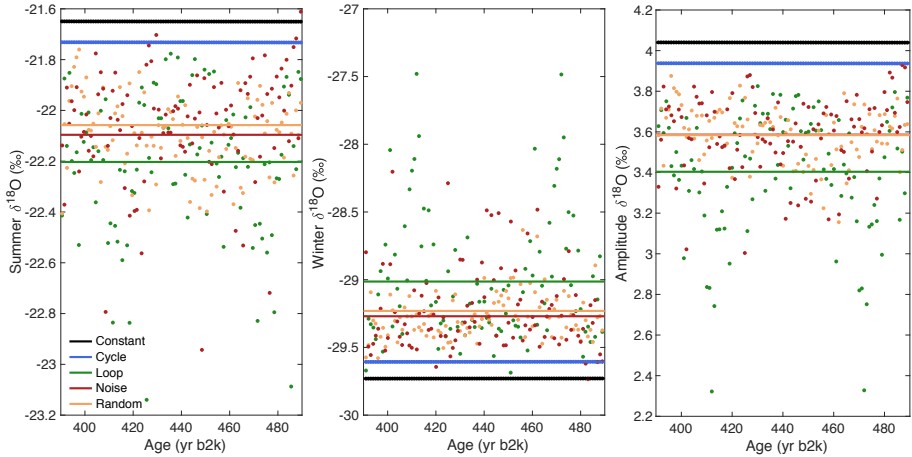

**Figure 9.** Comparison of diffusion-corrected seasonality for five different accumulation scenarios: constant, cycle, loop, noise, and random. Scattered points represent individual summer and winter maxima for the diffusion-corrected CFM output, and solid lines represent the mean of the full 100-year window. The comparison of (a) summer, (b) winter, and (c) annual amplitude demonstrates a slightly greater diffusion-correction effect on the winter signal between the different accumulation scenarios. There is a maximum decrease of 15.7% from the mean amplitude of the 'constant' scenario to the 'loop' scenario.

ice core sites (at least ∼50% less). In the model, all seasonally-biased accumulation scenarios exhibit an annual amplitude that is slightly under-corrected for diffusion, and is smaller than the pre-diffusion amplitude. The greatest effect occurs on winter months which receive less accumulation in the MAR reanalysis. Seasonally-biased accumulation scenarios can be compared to the constant accumulation scenario, in which each month receives the same amount of accumulation and the diffusion-corrected amplitude matches the pre-diffusion signal. In comparison to the constant accumulation scenario, there is a maximum 15.7%

decrease in the annual amplitude of diffusion-corrected isotope values for varied accumulation scenarios, which is a direct result of the bias in the diffusion-correction. Thus, in the unlikely case that accumulation shifted from a constant scenario to a seasonal-bias over the last 2.6 ka at Renland, we could expect up to a 15.7% offset from the true value of the annual amplitude. There is no indication this would have occurred in any reanalysis or model product of which we are aware. Furthermore, the annual amplitude in the observed RECAP $\delta^{18}$O record increases by approximately 50% over the last 2.6 ka, which

is substantially beyond what can reasonably be expected from changes in accumulation bias. While this analysis may include uncertainties in the MAR reanalysis data, it provides an end-member possibility for highly unlikely shifts in seasonality of accumulation, demonstrating that the effect on $\delta^{18}$O seasonality is minimal in comparison to observed trends.

### 3.2.2 Melt layers

Since the Renland site is subject to warm summer temperatures, we must also consider the possible effects of melt layers on the

seasonal signal. A summer melt event could cause surface snow with a relatively high $\delta^{18}$O signal (ie. near the seasonal peak) to percolate vertically through the firn column, mixing with the underlying winter layer which has a lower $\delta^{18}$O value. This

mixing would cause the preserved $\delta^{18}$O value of the winter layer to increase, resulting in a decrease in the annual amplitude. Alternatively, melt water which refreezes in the firn as an ice lens can produce a local barrier to further diffusion. In either case, it is important to consider the extent to which melt layers could influence the recorded water isotope signal, which we do so by examining both the isotope data and the melt layer density in the RECAP core.

Diffusion in a firn column without melt layers is expected to produce isotope data in which the fit to the diffused portion of the PSD is a Gaussian (Johnsen et al., 2000). Substantial alteration of firn processes due to melt, either through liquid water mixing or an ice lens barrier to diffusion, would likely influence the shape of the spectrum. Over the time period in which we reconstruct the annual signal, we do not observe degradation of the Gaussian fit in the 300-year windows of PSD, indicating that melt has not significantly influenced the isotope data. Additionally, it has been noted that melt layers can cause a 'ringing' effect in the diffusion-corrected data, resulting in spurious high-frequency oscillations (Vinther et al., 2010). We do not observe this effect in the diffusion-corrected isotope data at RECAP over the last 2.6 ka.

The density of melt layers at RECAP was measured by Taranczewski et al. (2019), determining a high-resolution record for the last 2.1 ka. The ratio of snow water equivalent of a melt layer to the respective annual layer is characterized as the annual melt ratio (AMR), which over the last 2.1 ka has an average value of less than 2% and is therefore a very small fraction of the total annual ice volume (<1 cm for 45 cm ice per year). The AMR exhibited some centennial variability with a few distinct periods of increased melt, but did not demonstrate a long-term trend over the last 2.1 ka (Taranczewski et al., 2019). Furthermore, during brief periods in which there is a ~1% increase in AMR (ie. 1850–1700 yr b2k, 200 yr b2k–present), increases in melt layer occurrence would likely serve to decrease the annual amplitude, which we do not observe. Based on this evidence, it is likely that the presence of melt layers is not significantly influencing the seasonality trends observed in the $\delta^{18}$O signal.

### 3.2.3   Insolation

We next consider the effects of insolation, which is expected to directly influence seasonal temperatures, and by extension the water isotopes observed in the RECAP core. To the first-order, orbital parameters (i.e. eccentricity, obliquity, precession) modulate the timing and intensity of the top-of-the-atmosphere (TOA) incoming radiation, and should drive peak summer temperatures and isotope values. Because Renland is located at a high latitude, the site receives minimal or no insolation throughout the winter, meaning that integrated summer insolation dominates. Thus, winter temperatures are less dependent on direct solar input, and instead are subject to the effects of lateral atmospheric heat transport and the efficiency of cooling from summer.

Here we assess the potential direct influence of changing solar insolation due to variations in the Earth's orbit on seasonal radiative equilibrium temperatures at Renland. We calculate the TOA insolation forcing at 71° N (Fig. 10a–c) (Huybers, 2011), which shows a small decline in maximum (and integrated) summer insolation over the last 3 ka, as well as a slight decline in annual mean insolation. We use a relatively simple energy balance model to calculate expected changes in seasonal surface temperature (Fig. 10d–f), that accounts for the above changes in insolation, temperature-dependent emission back to space, and horizontal atmospheric heat transport which is modeled as the diffusion of near-surface moist static energy (Hwang and

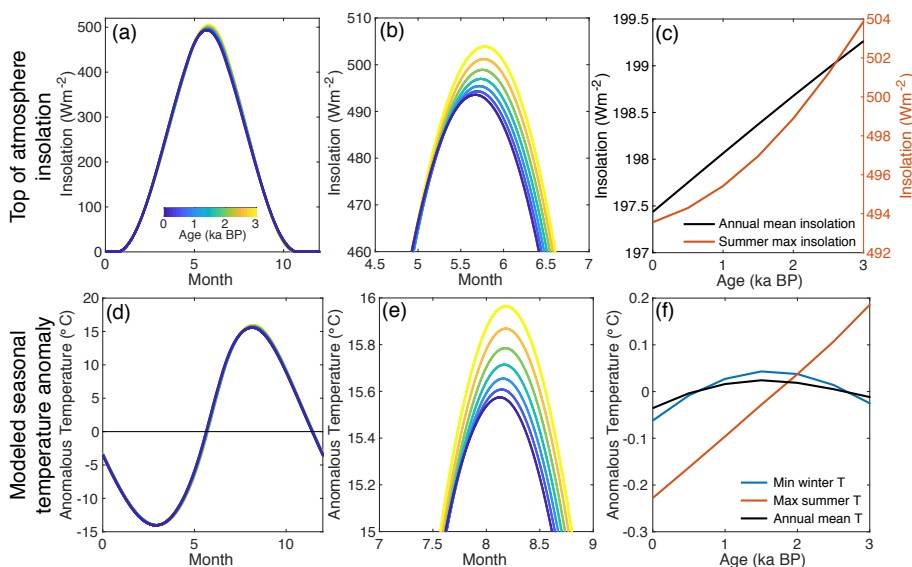

**Figure 10.** Changes in late Holocene insolation (a–c) and expectation for temperature change (d–f) from a simple energy balance model. (a) Seasonal cycle of insolation at 71° N, colored by age (in ka BP, years before CE 1950). (b) Same as (a) but zoomed in on summer. (c) Changes in maximum summer insolation (red) and annual mean insolation (black) at 71° N over the last 3 ka. (d) Modeled energy balance temperature anomaly at 71° N (as deviation from annual mean), colored by age (in ka BP). (e) Same as (d) but zoomed in on summer. (f) Changes in maximum summer temperature (red), minimum winter temperature (blue), and annual mean temperature (black) at 71° N.

Frierson, 2010). Expected peak summer temperatures decline slightly over the last 3 ka, in line with declining TOA summer insolation (Fig. 10c,f), while minimum winter temperatures remain relatively unchanged. These results are robust to a number of assumptions in the simple energy balance model. The RECAP seasonal $\delta^{18}$O signal shows a stable summer signal and decreasing winter signal; therefore, expected seasonal temperature trends due solely to changes in insolation are not able to
account for the observed changes in seasonal $\delta^{18}$O at the Renland site.

### 3.2.4    Regional climate variables

Indirect solar effects may influence other parts of the climate system, which then affect the local climate at Renland. A cooling trend in the North Atlantic is observed over the last 3 ka in records of glacial expansion, ice sheet growth, and increased drift ice (Miller et al., 2010), driven by a decrease in total annual insolation (Kaufman et al., 2009). In the RECAP core, a decrease
in $\delta^{18}$O is observed in winter, while the summer signal remains stable. While it is difficult to determine the cause of this trend without isotope-enabled modeling, we can hypothesize how possible mechanisms involving regional climate variability might influence seasonality of the isotope signal at Renland.

Since Renland is closer to the coast and open ocean than other inland ice cores, sea surface conditions could play a substantial role in Renland climatology (Holme et al., 2019). One possible factor is sea ice extent, which is both influenced by annual
insolation (Müller et al., 2012) and influences total absorbed insolation at the surface due to albedo. A number of studies have

documented an increase in sea ice cover in the North Atlantic and Fram Strait over the last 3 ka (Müller et al., 2012; Fisher et al., 2006; Jakobsson et al., 2010; Polyak et al., 2010). Sea ice extent has been previously studied through impurities in the RECAP core; iodine concentrations from the RECAP ice core suggest increasing sea ice over the last 3 ka (Corella et al., 2019; Saiz-Lopez et al., 2015), and bromine enrichment has been used to estimate sea ice conditions through 120 ka (Maffezzoli et al., 2019).

The formation of sea ice primarily occurs in winter, and increasing sea ice would be correlated with decreasing regional temperatures in winter and for portions of the shoulder seasons, depending on the timing of ice formation in fall and melt in late spring. A more open ocean regime at 2.6 ka, driven by higher total annual insolation, would keep winters warmer in coastal Greenland due to ocean heat contribution to the atmosphere (Screen and Simmonds, 2010). In recent centuries prior to the Industrial Revolution, lower total annual insolation and increased sea ice would dampen the moderating effect the open ocean has on coastal winter temperatures, resulting in colder winters. Increasing sea ice is therefore consistent with increasingly colder winters at Renland, whereas summers would largely be immune to sea ice response since nearby water bodies have little to no summer sea ice. This may explain the similarity between the winter $\delta^{18}$O signal and total annual insolation at 71° N (Figs. 10c, 8c).

However, it is nearly certain that other regional climate variables have an influence on the $\delta^{18}$O signal in the RECAP core. At this time we lack a comparison to seasonality at other locations in Greenland, which would help to determine the extent to which the trends observed in the RECAP record are due to local or regional influences. The RECAP core is unique in that it has both high sampling resolution (0.5 cm, whereas most other cores are over 2 cm), and high accumulation rate (45 cm yr$^{-1}$ compared to 10–20 cm yr$^{-1}$ inland), allowing for a much more accurate diffusion-correction of the seasonal isotope signal. At Renland, we may also observe the effects of atmospheric and oceanic circulation patterns and sea ice extent, which can control the influence of local oceanic moisture in comparison to long-range transport. This would alter the $\delta^{18}$O signal through moisture source instead of a direct influence on local temperature (Johnsen et al., 2001; Klein and Welker, 2016). As we do not have records of isotope seasonality from inland Greenland, it is difficult to identify whether an effect such as this uniquely influences the RECAP isotope signal. These factors could be instead be further explored through additional modeling studies.

## 4 Conclusions

The RECAP ice core from coastal East-Central Greenland contains a high-resolution water isotope record of the Holocene, obtained using continuous flow analysis (Jones et al., 2017b). The coastal proximity of the Renland ice cap makes the isotope record subject to influence from sea surface conditions (Holme et al., 2019). The record preserves annual variability for the last 2.6 ka, interannual variability for the last 5.5 ka, and decadal variability for the last 8 ka. We perform a diffusion correction calculation on the annual signal, based upon the diffusion length fitting routine of Jones et al. (2017a).

The diffusion correction calculation for annual variability requires testing by a firn model to determine if biases in seasonality of accumulation can alter the patterns observed in summer and winter extrema. Since diffusion corrections assume constant annual accumulation rates, diffusion correction can over- or under-estimate the summer and winter signals. We utilize the

Community Firn Model (CFM) (Stevens et al., 2020) to test an extreme case of shifts in seasonality of accumulation, from constant accumulation to summer-fall weighted accumulation, as given by MAR (Fettweis et al., 2017). We cannot rule out that centennial variations in summer and winter isotope values could be the result of seasonality of accumulation and the associated diffusional effects. However, the millennial-scale trend is robust: The summer extrema are relatively steady, and the winter extrema are steadily declining with a greater magnitude of change than can be accounted for by changes in seasonality of accumulation. We suggest that increasing sea ice driven by decreasing total annual insolation over the last 3 ka could cause the winter decline; however, other regional climate variables such as atmospheric and oceanic circulation may also exert an influence on the seasonal isotope signal. Future isotope-enabled modeling studies are needed to constrain how these factors are reflected in the RECAP $\delta^{18}$O signal.

The interannual and decadal signals are variable at millennial and centennial timescales, with a somewhat constant trend over the last 8 ka. This is to be expected, as the background climate state of the Holocene is also stable. We note that variability centered on 20 years has millennial-scale variations that appear similar to Bond Events recorded in a North Atlantic sediment core off the west coast of Iceland (VM 28-14) (Bond et al., 2001). The relationship is significant, but with low correlation for the full 8 ka record. However, VM 28-14 has large dating uncertainty, which could lessen the correlation. This potential relationship may be of interest for future isotope-enabled modeling, as this sort of millennial variability is pervasive in the North Atlantic (Bond et al., 2001).

Future ice core studies at locations with the annual signal preserved will provide additional constraints on spatial and temporal variability in the annual water isotope signal. At this time, Renland is the only available Greenland ice core that has high-resolution sampling and high accumulation rates. These factors are necessary to rule out seasonality of accumulation effects on diffusion, allowing for an interpretation of the summer and winter patterns for the last 2.6 ka. Whether Renland is unique in its downward trend in winter values remains to be seen. We suggest careful planning for future studies of the annual signal: 1) If site accumulation rate is low, impurity data will be needed to constrain the seasonality of accumulation; and 2) Isotope-enabled modeling should be used once new records are obtained, which will improve our understanding of regional climate dynamics.

*Data availability.* Holocene $\delta^{18}$O data through 10 ka will be made available on the PANGAEA data archive. Additionally, the $\delta^{18}$O record through 2 ka, including raw, diffusion-corrected, and peak detection results, will be submitted to the Iso2k database for public use in further analysis.

**Appendix A: Figures**

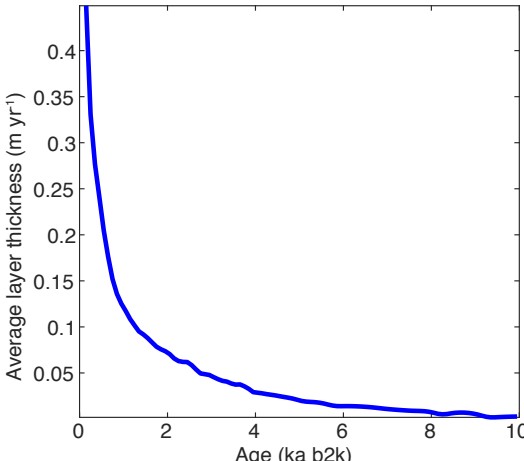

**Figure A1.** Mean annual layer thickness in the RECAP core throughout the Holocene; the ice cap is rapidly thinned over the last 2 ka.

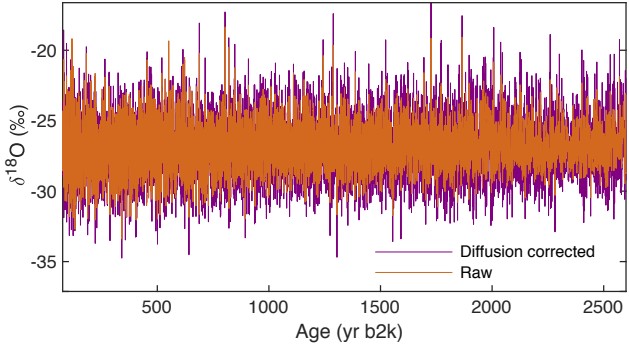

**Figure A2.** Raw (orange) and diffusion-corrected (purple) $\delta^{18}O$ for the period from 63–2600 yr b2k for which the annual signal can be diffusion-corrected.

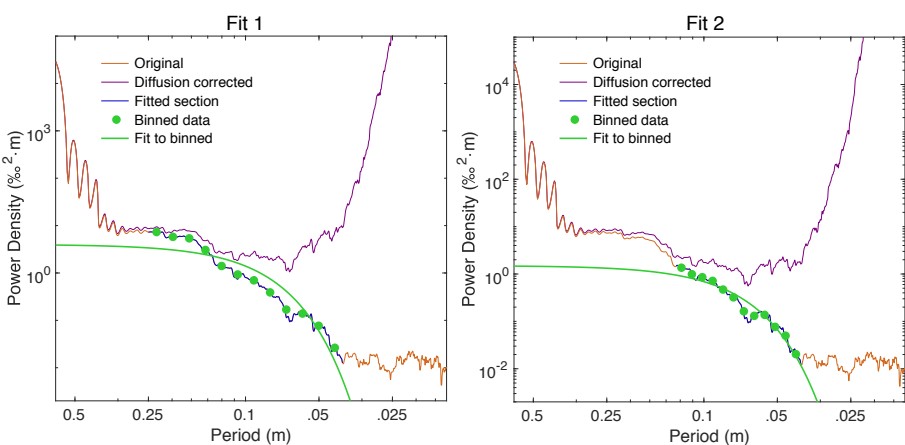

**Figure A3.** Gaussian fits (green line) can be found for the diffused part of the spectrum (green points) for most 300-year PSD windows, but there are two problematic sections from 5.6–6.7 and 7.8–8 ka. Two fitting schemes are used for these sections to ensure bias is not introduced, both shown here for a window from 6363–6663 yr b2k which exhibits a double-Gaussian shape. Fit 1 (left) utilizes a similar fitting range as surrounding windows to determine the diffusion length. The fitting range for each window is interpolated from the reliable ranges applied to the spectrum on either side of the problematic section. This results in a poor Gaussian fit, but a better estimate of the fitting range. Alternatively, the fitting range for Fit 2 (right) is selected so that the second Gaussian is used to determine the diffusion length. This results in a better Guassian fit, but erratic changes in the fitting ranges applied to different windows, which is not observed in the reliable spectra.

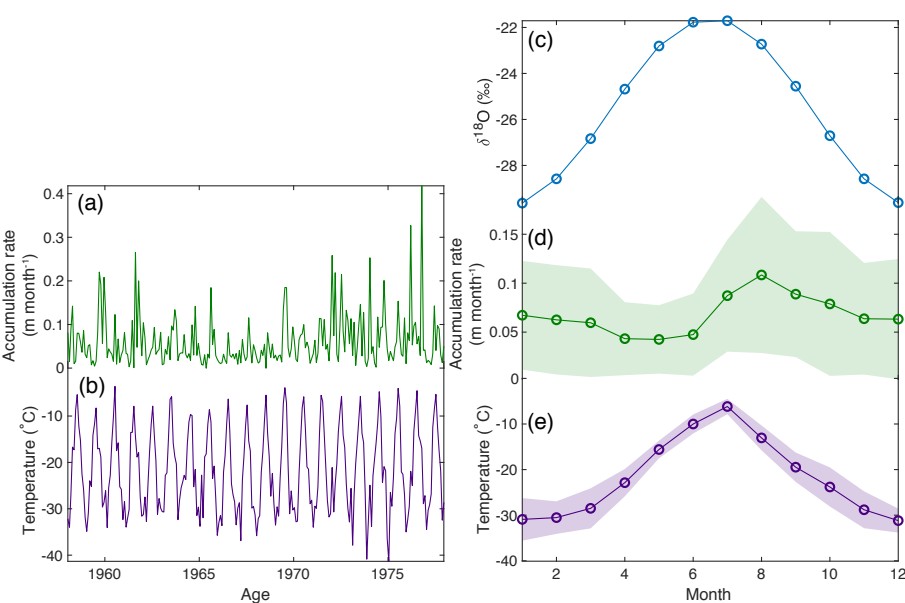

**Figure A4.** CFM input is based on the 20-year period of MAR data from 1958–1978, using (a) Monthly ice accumulation rate and (b) Temperature data. The monthly averages used for CFM input include: (c) Starting isotope data, based on the seasonal isotope signal observed at the top of the RECAP core; (d) Mean monthly accumulation rate from MAR; and (e) Mean monthly temperature from MAR. Shading indicates standard deviation from mean values from the 20-year period of MAR data. Renland experiences a slight bias for summer accumulation, with the months July–September receiving the most accumulation.

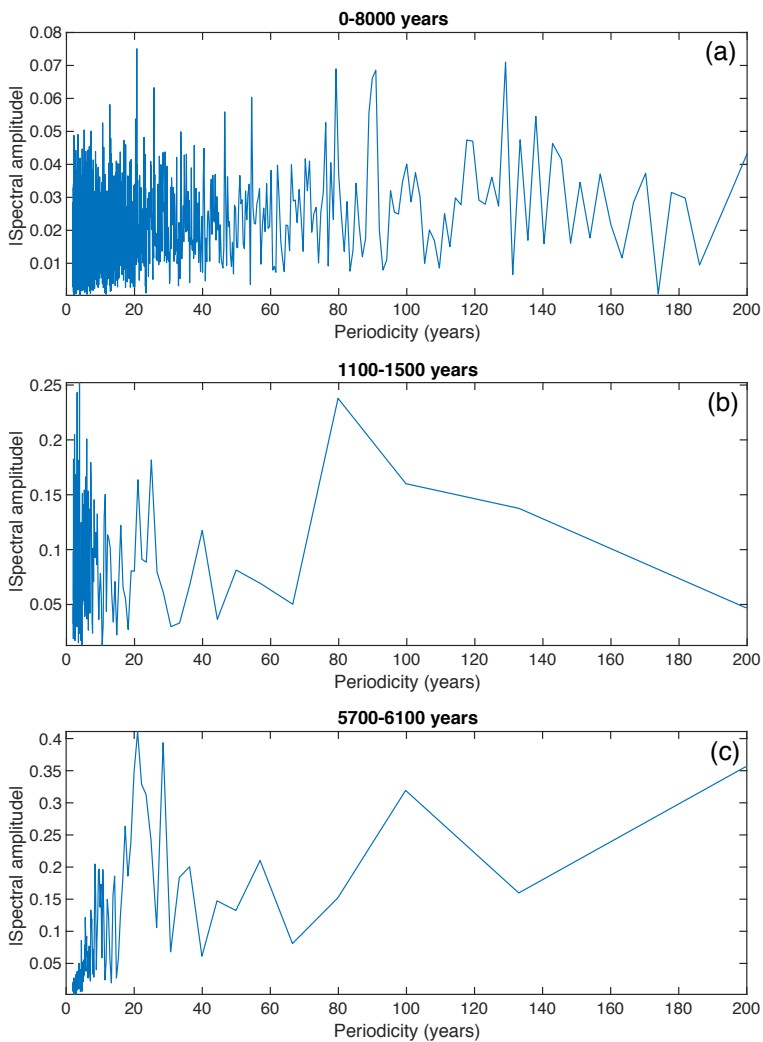

**Figure A5.** Examples of Fast Fourier Transform (FFT) of one-year smoothed $\delta^{18}$O demonstrate variability in the strength of the 20-year band, as reflected in the analysis of relative amplitude. (a) For the full time period from 0–8 ka, a small peak is observed at 20-year periodicity. (b) For the time period 1100–1500 years, the strength of the relative amplitude in the 15–20 and 20–30 year bands is low (see Fig. 6), and there is not a distinguishable peak at 20 year periodicity in the FFT. (c) From 5900–6100, there is a prominent peak in the relative amplitudes of the 15–20 and 20–30 year bands, and this is reflected in the FFT which shows a peak from approximately 15–30 years.

*Author contributions.* AH and TRJ contributed to all aspects of this paper. AH, TRJ, VG, BMV, VM, BHV, and CH contributed to processing of the RECAP ice core data. BMV developed diffusion-correction code, adapted by TRJ for this study. CS provided Community Firn Model output. BM provided insolation modeling. TRJ, AH, and JW developed and implemented analysis techniques. TRJ developed the extrema (summer, winter) picking algorithm. AH wrote the manuscript with significant editorial contributions from TRJ and comments from all authors.

*Competing interests.* The authors declare that they have no conflict of interest.

*Acknowledgements.* The RECAP ice coring effort was financed by the Danish Research Council through a Sapere Aude grant, the NSF through the Division of Polar Programs, the Alfred Wegener Institute, and the European Research Council under the European Community's Seventh Framework Programme (FP7/2007–2013)/through the Ice2Ice project and the Early Human Impact project (267696). The authors acknowledge the support of the Danish National Research Foundation through the Centre for Ice and Climate at the Niels Bohr Institute (Copenhagen, Denmark). Abigail Hughes also acknowledges support from the NSF through the Graduate Research Fellowship Program.

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
