# Peer review of "High-frequency climate variability in the Holocene from a coastal-dome ice core in East-Central Greenland"

_Climate of the Past, 2020_

## Referee Comment (RC1) · Anonymous Referee #1 · 25 Mar 2020

Review
**High-frequency climate oscillations in the Holocene from a coastal-dome ice core in east central Greenland**
Hughes et al, CPD, 2020

**General comments**
The authors present the d18O water isotope record for the Holocene from the RECAP ice core from Greenland. Using spectral analysis techniques, the authors compare the 15-20-year variability with the Bond Cycle. Using the Community Firn Model climate influences on diffusion correction are explored and a simple energy balance model is introduced to explore whether insolation and not sea ice could drive changes in d18O. Analysis of the seasonal signal of the last 2.6 ka reveal changes in the trends for summer and winter d18O signal. The authors speculate that these differences correspond to changes in sea ice conditions.

The manuscript is well written, and the methods and analysis are overall clearly explained. However, several places in the text, statements are written without showing sufficient values/analysis of the data to support these statements.

In the current version of the manuscript a large focus of the full manuscript is put on explaining the effects of sea ice variability on d18O signal in the ice core. This reflects an imbalance between the current well documented findings that the paper presents and the ideas and hypotheses that the authors mention without sufficient scientific argumentation.
There are reasons to suspect that the observed trends and correlations can be results of the post-processing of data and not directly an effect of sea ice. It is possible that sea ice is the driver of these effects but without clearly documenting (e.g. using a model or other proxy data) that sea ice is expected to influence the ice core site, the argumentation becomes a bit weak.
A clearer separation between method uncertainties and their resulting effects on one side and then a separate discussion on effects caused by climate/sea ice variability would strengthen the scientific argumentation significantly.

The presented d18O data from the Holocene part of the ice core is of great value to the scientific community, both on annual and seasonal values.
Connecting the RECAP ice core signal to the regional sea ice signal is a shared interest among paleo climatologists from several disciplines. It is therefore highly relevant that the authors pursue this connection. However, the authors are encouraged to significantly strengthen the analysis on method weaknesses regarding diffusion correction and strengthen the argumentation regarding the hypothesized sea ice influence on the d18O signal.

Based on the above I suggest publication with major revision.

**Major comments:**

Influence of diffusion correction on findings
In the paper by Vinther et al, 2010 (sec 4) the following statement is written "Looking at the 14 winter and summer season d$^{18}$O series presented in Figs. 5 and 6 it can be seen that the time series from Renland and DYE-3 show least variability in the high-frequency domain. It should be noted immediately that this apparent lack of variability is a consequence of the

particular diffusion correction applied to these series and should not be interpreted as a consequence of a different climatic forcing. «

This highlights an important caveat of this paper. Diffusion correction can influence variability in the signals as a result of the method itself. The effects of this must be clearly and thoroughly demonstrated. In addition, it is relevant that the authors clearly state how the method applied in this study differs from the method in Vinther et al 2010 and thus that the diffusion correction is ok to apply for RECAP.

When the strengths and weaknesses of the applied method are introduced it is meaningful to first thereafter explore effects of climatic variability on the diffusion correction, as done with the CFM model, which is introduced to explore the effect of changes in accumulation seasonality. Please also discuss issues regarding the interplay between accumulation seasonality, insolation and sea ice changes. Can accumulation seasonality in reality considered to be constant or are the combined effects on diffusion correction larger?

Sea ice signals in the RECAP core
A change in sea ice does not always directly change into a similar change in d18O. See e.g. Holme et al., 2019, Faber et al. 2017, Sime et al. 2013, Divine et al., 2011. And for paleoclimate signals on Merz et al., 2015 and Li et al 2010.
 The authors are currently not demonstrating the processes in which a regional sea ice change near RECAP translates into a changed d18O signal. Existing literature is used to argue that a link is plausible through d180, sea ice and AMOC, but the demonstration that this is actually the case for RECAP is missing.
Maffezzoli et al 2018 explored sea ice in the RECAP using impurities.
The findings from this paper is extremely relevant to include here in order to argue for how sea ice variability is "seen" from the RECAP core using impurities.
In the current approach the authors introduce a simple energy balance model to only because the variability in surface temperature (and therefore d180??) is not caused by insolation and thus indirectly argue that sea ice is the driver of the variability. This is not convincing. Effects on d18O and atmospheric circulation are not considered in this approach.
 I strongly suggest that the authors to include the  use  of (isotope) model simulations, moisture source tracking or similar to strengthen the argument on how the connection between sea ice and RECAP d18O variability must be used in order to demonstrate that  the effects of Holocene sea ice variability is  reflected in the d18O of RECAP.

RECAP and GRIP comparison (appendix A)
This is interesting and important.
The study argues that RECAP record signals of sea ice variability.
Thus, it is important to demonstrate that RECAP is unique in this sense and that the same variability patterns are not found in the same cores.
Differences in terms of accumulation and measurements resolution exists among the cores which creates issues, but the authors must present stronger arguments for why they find that RECAP variability is unique and driven by sea ice.
This deserves to be treated thoroughly in the manuscript instead of the appendix

Melt
The authors address the effect of melt in L264ff.
For the Holocene, frequent summer melt must be expected at RECAP.

It is unclear how melt is influenced the d180 seasonality trough vertical mixing of summer and winter layers. Given the importance of reconstructing a correct summer/winter d18O signal for the conclusions of this paper, the role of melt on seasonality is relevant to address further. The authors are free to find the best approach to address this challenging issue.

Sec 3 "Results and discussion".
I encourage the authors to separate the results from discussion to not mix up results from this study with hypotheses entirely based on other studies.

Comparison with VM28-18
It is relevant and meaningful to compare the data to this core and the authors argues for this in a good way. It is however symptomatic for this analysis that this hypothesis is entirely driven by other studies and the data and analysis in this study does not convincingly support this hypothesis (full span of the records r=0.34).
In the current format of the paper this link with the Bond cycles provide an argument for connecting d18O records to sea ice.
I suggest the authors to reconsider whether this approach is optimal.
If yes, then the analysis of the correlation between d18O and Bond cycles must be statistically stronger to ensure that this correlation is not just noise and coincidence.
If no, then please explore d18O and sea ice in alternative ways as explained later in this review.

Sec 2.5 Community Firn Model (CFM)
This section has several issues:
The motivation for introducing the CFM, is only mentioned in the end of the section and is unclear for the reader. Please expand the text to explain accordingly.

Using CFM with input from MAR forced by ERA-Interim creates a long chain of uncertainties and known model biases. Especially for the Renland Ice Cap which is not represented well in most model grids. Nearby coastal and AWS weather stations exists, please demonstrate that the model results are in line with nearby observations.
The outcome/results of this method are not clear from the text (but clearer when fig 10 is shown)

Methods:
Please add a short separate section for information regarding the location and drilling of the RECAP core (see similar in Holme et al, 2019 and Maffezzoli et al 2018)

Naming RECAP vs Renland
The manuscript refers to the ice core as Renland (except for the Figure caption of fig 2)
The Renland core was drilled in 1988 and RECAP (REnland ice CAP) ice core
in 2015 nearby, both on the Renland Ice Cap. I suggest that the text is corrected in a suitable way to avoid misunderstandings and reflect the correct naming of the core.

Comparison with existing record (Renland 1988)
How does the former Renland core compare to the newer RECAP core? Do we see the same signal for the Holocene period or are there any deviations? Other than differences in instruments and precision the cores are not expected to be very different, but it would be good to demonstrate the agreement with the previous core. This can for instance be done in one of the very first figures.

Figures:
The manuscript is relatively long and have several figures of less relevance. It would strengthen the quality and readability of the manuscript if some of these less relevant figures would be placed in the supplementary instead. Examples are Fig 5, Fig 6,7

**Minor comments**

Figure 1:
Please add the description of the location markers and the reasons for the different colors (ice, marine etc) to the figure caption

L29:
This reference "Noone and Simmonds 2004" concerns conditions in Antarctica and is not suitable here. Please use a better reference, e.g. evidence from other analysis on the same core or similar (e.g. Holme 2018)

L29-30: "These climate parameters are recorded in the ice core water isotope (i.e. $\delta$ 18 30 O) record through changes in condensation temperature at the time of precipitation». This statement does not agree with the last 10-15 years of isotope and ice core research. Please add a few lines with supporting newer references that explain how the ice core isotopes is an integrated signal of several processes from source to site.

L60-65: This should be well known to most readers, and is not relevant in the method section

L88-98: This text is mainly "textbook material" and does not belong in a method section. Please shorten this.

L164. Please argue for the choice and robustness of assuming a 4 permill sine wave for the amplitude which is larger than shown later.

L163: The authors chose to force the MAR model with re-analysis data in a time period pre-satelite era. Biases in this time period is expected to be large in the Artic region given the very limited data. Please discuss, maybe based on Fettweiss et al 2017, whether this choice is expected to introduce new biases to the results.

L165: exchange the word predict with simulate. (The model simulates temperatures in the past)

L165: "… July-September receive the most precipitation on average". How much more precipitation comes during summer than winter? Please add the relevant numbers to support this statement including relevant statistics as precipitation is highly variable on Greenland, especially on the coast.

L167-172. Please be clear how and if the sum of annual accumulation varies from year to year in each of the scenarios.

Fig 5,6,7: I suggest that these figures are added to the supplementary material instead.

Figure 5: Please add the standard deviations to the figure. The figure B3 demonstrate large variability.

L191:
Could the variability in correlation strength be due to methodology/age model?
Are there reasons to believe that age model differences in the Bond core can explain this mismatch? Please discuss this.

L192: Please clarify that the p-values are calculated on time series without significant autocorrelation

L214 If this is a finding from Holme et al 2019, please connect this statement with a repeated reference to this paper.

Fig 9: is panel (a) necessary to plot? Removing this would make this figure a little less chaotic.

L250:
"The model results show that the Renland diffusion correction is minimally influenced by seasonality of accumulation (Fig. 10)". How is this clear? Please argue with numbers and a changed design of fig 10 (see below)

Fig 10: This plot is difficult to read and interpret. Either plot these on two different plots for summer and winter, or consider plotting the anomaly from the mean instead.
In the current version of the figure it is unclear what the authors wants to demonstrate.

Figure A1: Please improve the figure caption to separately describe the left and right panel

Appendix A1
It is argued that the effect of sea ice on GRIP is muted, but Fig A1 right panel show that both experience an increase in amplitude, but GRIP looks muted as the range of d180 is larger for GRIP. Please plot these on comparable scales. And argue sufficiently that RECAP is unique compared to GRIP.
 The conclusions about the comparisons in this section are currently not fully supported by values from analysis of the used models and data.
E.g.
L342: "A caveat in analyzing this data is that accumulation may have a greater seasonal bias at GRIP".
Please support all statements in the appendix with values e.g. from MAR, CFM and ice cores.

L335: Following open access style please provide all data types (raw and formatted) online before publication.

---

## Referee Comment (RC2) · Anonymous Referee #2 · 7 Apr 2020

Review of manuscript cp-2020-19:
"High-frequency climate oscillations in the Holocene
from a coastal-dome ice core in east central Greenland"
by Abigail Hughes et al.

7th April 2020

Hughes et al. present a new oxygen isotope temperature proxy data set covering the Holocene from an ice core drilled on the Renland Peninsula ice cap in eastern coastal Greenland. The isotope record is obtained on a high nominal resolution from Continuous Flow Analysis coupled to Cavity Ring-Down Spectrometers. Originating from this high-resolution data set, the authors use spectral analyses to estimate diffusion lengths and to investigate the extent to which the isotope (climate) signal of certain frequencies and frequency bands survives the diffusional smoothing throughout the Holocene. By means of deconvolution, they reconstruct the seasonal summer and winter isotopic time series. Subsequently, the authors discuss the relationship of the isotope records' decadal variability with regional sea ice and ocean circulation changes as well as potential causes of the observed decrease in winter isotope values over the last 2.6 ka.

The paper presents a new proxy data set at high resolution and discusses its relevance to assess regional climate information. This is an important contribution since the interpretation of isotopic data from coastal ice cores is an up-to-date topic and because the investigated site has only been studied previously at relatively low temporal resolution. The paper thus fits well within the scope of Climate of the Past. I would like to congratulate the authors on their elaborate analysis of the isotope data; the methods are sound, the paper is well written and structured and the figures overall adequately present and support the findings of the paper. However, my major concern with this work pertains to the interpretation of the results in terms of "high-frequency climate oscillations" and regional oceanic conditions (sea ice, ocean circulation). In the current state, I see only relatively weak evidence in the presented data supporting the claims made in title and abstract. Thus, I would recommend publication of the paper after a major revision of these general issues, which I outline below. In addition, I give a list of specific as well as technical comments you might want to address.

**General comments**

A first major point is that the main conclusions the authors give split up in two seemingly separate issues, i.e. the relationship of the decadal variability with sea ice conditions and/or ocean circulation, and the explanation of a decreasing winter trend by increasing sea ice. These two interpretations are presented rather disconnected throughout the manuscript, partly due to the combined structure of results and discussion, but also in the final overall conclusions section. As a result, the reader is somewhat left puzzling what the overall essence of this paper is, how the two presented main results connect with each other and what we in general learn about interpreting isotope records from coastal regions.

This problem is also reinforced by the presented evidence to support the interpretation, which is rather weak in my opinion.

The title of the paper alludes to observed climate oscillations with a high frequency. Firstly, I would not denote decadal variations as being high frequency, but more importantly, the authors present a

rather simplistic and too deterministic view on climate variability. Strictly speaking, an oscillation is defined as a periodic variation in time with a fixed frequency, or in a coupled system, a superposition of several frequency modes. For climate variability, this deterministic view is not necessarily the case. The Atlantic Multidecadal Oscillation might indeed appear to be a quasioscillatory phenomenen, but very likely it does not exhibit a strict 20-year cyclicity, as suggested in the paper (P3 L45). By contrast, the spectrum of the NAO is essentially white, so these are rather random variations than "subdecadal climate [or] multi-year cycles" (P3 L46-48). This also applies to the "Bond cycles". There seems to be no single deterministic cause (Wanner and Bütikofer, 2008), the 1500-yr periodicity is likely an artifact (Obrochta et al., 2012), and the variations could be simply compatible with internal climate background variability (see also the discussion of the statistics of D-O events as their glacial counterparts; e.g. Lohmann and Ditlevsen (2018)). I would suggest that the authors critically revise the text passages where these phenomena are discussed; especially focussing on toning down statements of Bond events having a 1500-yr periodicity (P2 L35, P11 L208, P17 L318) an rewriting the respective introductory paragraph (P2 L35 – P3 L42). Here, maybe use AMOC as a starting point and then develop its influence on local climate and sea ice?

The comparison of the 15–20 year variability with the Bond curve (Figs. 7 and 8) is also critical in this regard. While the title of the paper suggests that you observe oscillations in the isotopic time series, what the authors actually show in the figures here is the time evolution of the isotopic variability (if I understand correctly, either the average spectral power or the variance; see specific comments) in the frequency bands. While this is certainly a very interesting analysis, the difference to "normal" variations in the time series should be made clearer to the reader in the text and the title of the paper. Moreover, a discussion of a possible mechanism would be welcome: what could link temporal changes in drift ice to changes in isotopic variability, i.e. non-stationarity, within the 15–20 year frequency band? Alternatively, what is the significance of the curves in Figs. 7 and 8? Are the variations maybe just within the curve's uncertainty range? An error shading including spectral estimation error and diffusion-correction uncertainty would be helpful. From Fig. 7, I would say that visually the 15–20 year band exhibits correlation with the 20–30 year band and to a lesser extent also with the 7–15 year band; can you comment on this? This could either arise from correlation in the spectral uncertainty, or the speculated link to climate is not confined to the 15–20 year band. Stacks of Greenland ice-core records show stronger variability compared to the background around the 20-year period in general (Chylek et al., 2011), so a comparison with the full RECAP isotope spectrum (either a diffusion-corrected or, if this is tedious, the raw one) would help to better place the new spectral data into context with existing data from the region.

Finally, I have some concerns regarding the interpretation of the seasonal isotope data. In section 3.2.3 the authors conclude that the decreasing winter trend in isotope data could be explained by increasing sea ice which correlates with decreasing winter temperatures. In this regard, the authors also cite the paper by Noone and Simmonds (2004). However, as far as I understand this study, the effect of changing sea ice on isotopic composition is rather through controlling the influence of local oceanic moisture in comparison to long-range transport, instead of a direct influence on local temperature. It would thus be worth to discuss the influence of sea ice on isotopic composition in more detail here.

Furthermore, the authors also state as a main conclusion that this finding "is a valuable demonstration of the additional regional climate information contained in coastal ice cores" (P18 LL328-329). However, I would be more convinced if you could explicitly show that the RECAP spectral (see above) and seasonal signals are clearly distinct from other Greenland ice cores. I don't see this being convincingly presented. The GRIP seasonal data basically shows the same features (stable summer, decreasing winter; cf. Appendix A) – isn't the fact that the resulting increase in GRIP seasonal amplitude appears "muted" compared to Renland (Fig. A1) simply explained by the difference in axis scaling ($\sim$ 1.5‰ axis range for RECAP compared to $\sim$ 3‰ for GRIP)?

**Specific comments**

– P1 L7: "The strength of the interannual frequency band decays rapidly". I suggest to remove this result, since it is a direct consequence of firn diffusion and therefore a trivial and expected result for almost any ice core, which is not relevant for the abstract.

– P1 LL8-9: "Comparison to other North Atlantic proxy records suggests": As far as I follow the paper, the results concerning the Renland 15–20 yr variability are compared to only one other proxy record, which is the VM28-14 record of Bond et al. (2001), so this statement should be adjusted.

– P1 L17: "Greenland ice core records are valuable for determining a more comprehensive picture"; more comprehensive compared to what or which other records? This is not clear from the context provided.

– P1 L20: "[...] were used in analysis of the Renland ice core". This is a bit confusing since there is also the old Renland ice core which was already drilled in 1988. If I am not mistaken you present and refer here to the new RECAP ice core. Please clarify this throughout the paper.

– P2 Fig. 1: Please introduce the scope of the figure in a first sentence, e.g. "Map of the study region", and add to the caption that the red circle denotes the RECAP (?) drill site, blue circles other Greenlandic ice core sites, and the green circle the location of a marine sediment drift ice proxy record.

– P2 LL28-29: The Noone and Simmonds (2004) paper explicitly investigates the influence of sea-ice cover on western Antarctic ice cores; the statement here would suggest to a reader unfamiliar with the topic that it instead covers the link between coastal Greenland climate and ocean conditions, which is not the case. Please adjust the statement, stating that a similar influence of sea ice as observed in Antarctica could be expected for coastal Greenland cores, or provide a directly relevant reference. This similarly applies to the same reference in Sect. 3.2.3 (P17 L301).

– P1 L21 – P2 L34: The flow of information is rather incoherent in this paragraph. Please consider rewriting it such that you start to introduce the new RECAP core and then describe the peninsula, the ice cap and the local climate.

– P3 LL69-70: "due to a small amount of mixing introduced in the system". Please explain the CFA mixing effect shortly here and either state the amount or provide a reference.

– P4 L79 – P5 L86: As I understand it, the main reason for the loss in high-frequency variability is thus in both cases diffusion (as expected), but amplified by two distinct reasons. Please restructure the paragraph accordingly to highlight this and mention diffusion as the cause in the first place, and then elaborate the two reasons for the strong diffusion within the LGM and the basal ice.

– P5 LL90-91: "along concentration, temperature, and vapor-pressure gradients"; I would argue that concentration and vapor pressure are directly linked, so stating both is redundant. Just mention concentration, as it is more intuitive.

– P5 L104: Please provide a reference for the CFA system mixing effect, if not done before (please see respective comment above).

– P5 LL105-106: "with a 100-year time step"; at first reading, this can be misunderstood. I understand you mean that the step size between the overlapping windows is 100 years, but as it is written it might be confused with the temporal resolution; please clarify. Please also underline that you produce a spectrum for each window. Additionally, it is more common to refer to this quantity as the power spectral density (PSD); please change this throughout the manuscript.

– P5 L111 – P6 L118: Why is the linear regression in log space only used to assess the uncertainty of the diffusion length but not to estimate the diffusion length value itself? Should this not maybe provide a better fit than a nonlinear model? Or do you assume any other shape for $P_0(f)$ than white noise for the fit? What about the measurement noise – is it subtracted before? You do talk about this some point later in the manuscript, but it should be mentioned here already.

– P6 Eq. 3: I am not sure if I understand your uncertainty estimation correctly. From the slope $m$ of the regression of $\ln(P)$ against $f^2$, one obtains the diffusion length as $\sigma = \sqrt{m}/(2\pi)$. If the slope estimate has some uncertainty $\Delta m$, then the uncertainty of the diffusion length should be $\Delta\sigma \sim \frac{\partial\sigma}{\partial m}\Delta m = \frac{1}{2\pi}\frac{1}{2\sqrt{m}}\Delta m$; so the exponent in Eq. (3) seems to be wrong ($-\frac{1}{2}$ instead of $\frac{1}{2}$), and isn't the slope uncertainty missing in there as well?

– P6 L124-130: Please clarify; it is clear that the diffusion length in depth should decrease due to thinning, but for the sake of clarity it could be worth to mention that, to a first approximation, the firn diffusion length should stay constant in the time domain after pore close-off, so an observed increase in diffusion length in time must have a different cause, i.e. ice diffusion etc.

– P6 L134: There is not really much information regarding the two fitting methods to be found in the appendix or the respective figure caption.

– P6 L139: What about the contribution of non-climatic noise (e.g. from stratigraphic noise; Münch and Laepple (2018)) to the spectra?

– P6 L140: I would not call it amplitude at all, since this term refers to truly oscillatory behaviour. Just use "strength" or "strength of the variations". Also it is not clear to me which exact quantity you investigate: do you use the square root of the power at the respective frequencies or averaged over the frequency band? Or do you integrate (for the frequency bands) the power spectrum to obtain the average variance (and then standard deviation from the square root of it) within each band? Please clarify here and also when discussing the results in Sec. 3.1.

– P6 LL141-142: "individual frequencies are normalized", "each frequency band is normalized"; better write that not the frequencies are normalized but the PSD/variance at the frequency/within the frequency band.

– P7 Fig. 3: Please introduce the scope of the figure in a first sentence, e.g. "Isotope power spectra and diffusion length estimation." Also, can you comment on the rather unusual shape (sharp increase, then flat) of the estimated spectra in panel (a) for large periods ($> 3\,\mathrm{m}$); which part of it can be trusted and which might be an artifact of the estimation method? In addition, it might be also worth to mention that the strong increase in PSD of the diffusion-corrected spectrum for large frequencies (small periods; violet curve) arises from blowing up of the measurement/CFA noise by the diffusion correction.

– P8 L154-156 and Fig. 4: I wonder, given the sub-seasonal variability of the raw (diffused) signal, if the smoothness of the deconvolved signal is an artifact of the inversion method and not real? In other words, would we really expect the original (pre-diffusion) seasonal $\delta^{18}$O signal to be so smooth? Have you tried to diffuse again your deconvolved signal and compare it to the original?

– P8 L164: "and a constant amplitude (4‰)"; why do you choose the amplitude about twice as large as observed from the seasonal isotope data (Fig. 9e)?

– P8 L165: One cannot really see the precipitation bias clearly on Fig. B3; better refer to Fig. 5 here and reference Fig. B3 in the first sentence. Also please consider to list the input information in a more coherent way: Start with the MAR temperature data, then mention precipitation bias in MAR and then your assumed accumulation scenarios.

- P10 Fig. 6: Please introduce the scope of the figure in a first sentence. Additionally, dissipation in a physical sense refers to the energy loss by friction; please rephrase, e.g. "the annual signal strength decreases"/"the annual signal diffuses".

- P10 L185 and Fig. 6 and 7: Why does the normalized amplitude has a unit (‰)?

- P10 LL186-187: Remove the sentence "Each frequency band is...", since this information is already given in the Methods section and the figure caption.

- P12 L219: What kind of "similar 20-year signal"? Between the time series (correlation of e.g. 20-yr averages) or in spectral properties?

- P13 L227: "of this relationship"; this is unclear: of which relationship? How does this sentence relate to the statement from the Knight et al. paper directly before?

- P13 L236: Please clarify what you mean with non-stationarity here. You just presented that the winter time series shows a trend, so is non-stationary in the the sense that its mean value is decreasing, but you say that the summer value does not show any clear trend, so it would be stationary. Or do you refer in both cases to other statistical quantities which show time dependence, e.g. the time series variance?

- P13 Sect. 3.2.1: This is a well-written section that convincingly argues that accumulation seasonality and changes in melt layers very likely are not the cause for the observed trends in isotope seasonality. The only thing which comes to my mind is, however, the possibility of changes in the seasonality in firn temperature to maybe affect the isotope seasonality via seasonally varying diffusion lengths (Simonsen et al., 2011). Could you provide arguments to constrain this possibility?

- P13 LL253-254: How can I see the pre-diffusion amplitude in Fig. 10 to assess the under-correction? Is it identical to the constant accumulation case diffusion-corrected value? Or does this case also lead to an amplitude under-correction?

- P14 Fig. 9: Please introduce the scope of the figure, e.g. "RECAP annual and seasonal $\delta^{18}$O data."

- P15 Fig. 10: Please clarify: "Comparison of diffusion-corrected seasonality...".

- P15 L255: I cannot see an 18% difference in Fig. 10; when I compare the green dots to the black line, there is at most a difference of maybe 7%?

- P16 LL274-275: "and the snow layers are so thick at Renland..."; this is repitition from above, you might consider removing this part.

- P17 LL291-292: It might be worth to elaborate a bit more on this conclusion. Do you mean that the insolation cannot explain the seasonal isotope changes since the summer insolation change is too weak or because there is no winter change, or both? It could also be useful to underline the importance of having seasonal information available due to the accumulation bias in the annual data. How do your results connect to the millennial-scale trends seen over the Holocene in other Greenland ice cores (Vinther et al., 2009)?

- P17 L307: Someone not familiar with the sea-ice edge distribution might wonder whether the area is actually ice free in summer at present?

- P17 L308: The first part of this paragraph seems to be a repetition from the previous one; please consider to restructure the two paragraphs.

- P18 L322 and L324: Given the speculative nature of your conclusions, I would rather nuance this and instead write "...and diffusion correction reveals a decreasing trend..." as well as "We instead suggest that the winter trend is likely...".

– P20 Fig. B2: I must admit that I do not really understand the difference in the fitting approaches from the figure caption or the figure itself. Please explain this in more detail.

**Technical comments**

– P1 L1 and title: Please capitalize and hyphenate consistently "East-Central Greenland" throughout the manuscript.

– P1 L4: replace "and the annual..." with "while the annual..."

– P1 L9: Please follow the house standard and use a long (em-) dash for range of numbers, so 15–20 instead of 15-20 (here and throughout the manuscript).

– P1 L16: Please hyphenate phrases such as "Ice-core records" throughout the manuscript; I will mention only some additional instances in the following.

– P1 18-19: Please change to "Cavity Ring-Down Spectroscopy".

– P1 L19: Please hyphenate "high-frequency signals" (throughout the text); see previous comment.

– P1 L21: Please change to "Renland Peninsula".

– P2 Fig. 1: Please change to "east coast".

– P3 L41: Please change to "northward".

– P3 LL49-50: Please change to "tree-ring data" and "central and western Greenland ice cores".

– P3 LL59-60 and throughout the manuscript: I very much appreciate that you introduce the correct terminology "water isotopologues" here. However, then please be consistent and avoid phrases involving "water isotope"; instead, simply use "isotope" (e.g. isotope signal, isotope record) or use "water isotopologue" or "isotopic composition", where appropriate.

– P3 L62 and Eq. (1): Change "SMOW" to "VSMOW".

– P3 L65: Equations are considered to be a part of the sentence, so please start this sentence with lower case "such" (also for Eqs. (2–4)).

– P3 L67: Change cm/min to cm min$^{-1}$.

– P3 L68: For the sake of clarity, please rephrase the sentence to "using two Cavity Ring-Down Spectrometers running in parallel (L2140-i and L2130-i)".

– P4 L74 "0–4035 yr bp": I guess you mean years before present (1950) here; please define this at the first instance and use "yr BP".

– P4 L79: Please change to "exibits a much higher effective resolution".

– P5 L97 "804.3 kg/m$^3$": there is no need to use such a precise number here (since it is an estimated value); please write "~ 804 kg m$^{-3}$".

– P5 L102-103: I think one most relevant reference to introduce the diffusion length would be sufficient (e.g. the first paper which introduced the concept in the context of firn diffusion).

– P5 L105: Change to "in the depth domain".

– P5 L106: Change "cycles/m" to "m$^{-1}$".

– P5 L108: Change to "To estimate the diffusion length" or "To estimate diffusion lengths".

- P5 Eq. (2) and P6 L115: Use upright font for mathematical functions such as "exp" and "ln".

- P6 L115: Change to "of the data".

- P6 L124: Change to "m yr$^{-1}$".

- P8 LL159-160: For the sake of clarity, please insert "to the resulting summer and winter time series" after "is applied".

- P9 L177: Change to "diffusion-correct".

- P10 L181: For the sake of clarity, please insert "with time" after "decays rapidly".

- P10 L182: Change to "diffusion-corrected".

- P10 L186: For what is the Jones et al. reference needed here?

- P11 Fig. 7 and P12 Fig. 8: The lines of the two fits (solid purple and solid dashed) are indistinguishable; please use another color for highlighting.

- P13 L243: Change to "diffusion-correct".

- P13 L249 and P15 L256: Change to "diffusion-corrected".

- P15 L259 "in which we are aware": I am not familiar with this phrase; do you mean "which we are aware of"?

- P16 Fig. 11 caption: Please clarify: "Changes in Holocene insolation..."

- P16 L279: Change "of top of atmosphere" to "of the top-of-the-atmosphere".

- P17 L305: Change Fall->fall and Spring->spring.

- P17 L316 "The Renland ice core": Please clarify "The RECAP ice core".

- P19 Figure A1 caption: Please rephrase: "Comparison of Renland and GRIP seasonal data. GRIP experiences...".

- P20 Figure B1 caption: Please clarify: "Mean annual layer thickness in the RECAP core...".

- P20 Figure B2 caption, second line: "to determine a diffusion": Do you mean "to determine the diffusion length"?

**References**

Chylek, P., Folland, C. K., Dijkstra, H. A., Lesins, G. and Dubey, M. K.: Ice-core data evidence for a prominent near 20 year time-scale of the Atlantic Multidecadal Oscillation, Geophysical Research Letters, **38** (13), L13704, DOI: 10.1029/2011GL047501, 2011.

Lohmann, J. and Ditlevsen, P. D.: Random and externally controlled occurrences of Dansgaard–Oeschger events, Clim. Past, **14** (5), 609–617, DOI: 10.5194/cp-14-609-2018, 2018.

Münch, T. and Laepple, T.: What climate signal is contained in decadal- to centennial-scale isotope variations from Antarctic ice cores?, Clim. Past, **14** (12), 2053–2070, DOI: 10.5194/cp-14-2053-2018, 2018.

Obrochta, S. P., Miyahara, H., Yokoyama, Y. and Crowley, T. J.: A re-examination of evidence for the North Atlantic "1500-year cycle" at Site 609, Quat. Sci. Rev., **55**, 23–33, DOI: 10.1016/j.quascirev.2012.08.008, 2012.

Simonsen, S. B., Johnsen, S. J., Popp, T. J., Vinther, B. M., Gkinis, V. and Steen-Larsen, H. C.: Past surface temperatures at the NorthGRIP drill site from the difference in firn diffusion of water isotopes, Clim. Past, **7** (4), 1327–1335, DOI: 10.5194/cp-7-1327-2011, 2011.

Vinther, B. M., Buchardt, S. L., Clausen, H. B., Dahl-Jensen, D., Johnsen, S. J., Fisher, D. A., Koerner, R. M., Raynaud, D., Lipenkov, V., Andersen, K. K., Blunier, T., Rasmussen, S. O., Steffensen, J. P. and Svensson, A. M.: Holocene thinning of the Greenland ice sheet, Nature, **461** (7262), 385–388, DOI: 10.1038/nature08355, 2009.

Wanner, H. and Bütikofer, H.: Holocene bond cycles: Real or imaginary?, Geografie-Sbornik, **113** (4), 338–350, 2008.

---

## Author Comment (AC1) · 20 May 2020

**General comments**

The authors present the d18O water isotope record for the Holocene from the RECAP ice core from Greenland. Using spectral analysis techniques, the authors compare the 15-20-year variability with the Bond Cycle. Using the Community Firn Model climate influences on diffusion correction are explored and a simple energy balance model is introduced to explore whether insolation and not sea ice could drive changes in d18O. Analysis of the seasonal signal of the last 2.6 ka reveal changes in the trends for summer and winter d18O signal. The authors speculate that these differences correspond to changes in sea ice conditions.

The manuscript is well written, and the methods and analysis are overall clearly explained. However, several places in the text, statements are written without showing sufficient values/analysis of the data to support these statements.

In the current version of the manuscript a large focus of the full manuscript is put on explaining the effects of sea ice variability on d18O signal in the ice core. This reflects an imbalance between the current well documented findings that the paper presents and the ideas and hypotheses that the authors mention without sufficient scientific argumentation. There are reasons to suspect that the observed trends and correlations can be results of the post-processing of data and not directly an effect of sea ice. It is possible that sea ice is the driver of these effects but without clearly documenting (e.g. using a model or other proxy data) that sea ice is expected to influence the ice core site, the argumentation becomes a bit weak.

A clearer separation between method uncertainties and their resulting effects on one side and then a separate discussion on effects caused by climate/sea ice variability would strengthen the scientific argumentation significantly.

The presented d18O data from the Holocene part of the ice core is of great value to the scientific community, both on annual and seasonal values. Connecting the RECAP ice core signal to the regional sea ice signal is a shared interest among paleo climatologists from several disciplines. It is therefore highly relevant that the authors pursue this connection. However, the authors are encouraged to significantly strengthen the analysis on method weaknesses regarding diffusion correction and strengthen the argumentation regarding the hypothesized sea ice influence on the d18O signal.

Based on the above I suggest publication with major revision.

We thank the reviewer for taking the time to provide a thorough analysis with constructive criticism. Below we individually address each comment, outlining changes we have made to improve the manuscript. All line numbers are in reference to the original manuscript.

**Major comments:**

Influence of diffusion correction on findings: In the paper by Vinther et al, 2010 (sec 4) the following statement is written "Looking at the 14 winter and summer season d18O series presented in Figs. 5 and 6 it can be seen that the time series from Renland and DYE-3 show least variability in the high-frequency domain. It should be noted immediately that this apparent lack of variability is a consequence of the particular diffusion correction applied to these series and should not be interpreted as a consequence of a different climatic forcing."

This highlights an important caveat of this paper. Diffusion correction can influence variability in the signals as a result of the method itself. The effects of this must be clearly and thoroughly demonstrated. In addition, it is relevant that the authors clearly state how the method applied in this study differs from the method in Vinther et al 2010 and thus that the diffusion correction is ok to apply for RECAP.

There are generally two strategies to account for diffusion in water isotope records of ice cores: 1) Back diffusion (estimated directly from the data) and 2) Forward diffusion (estimated from models). In case 1, as we have done in this paper, we correct for the effects of diffusion to recover an estimate of the isotopic signal as it would have existed at the surface of the ice sheet. In case 2, we can advance diffusion to be equal to the maximum amount of diffusion in the ice core; this will eliminate high-frequency information. Vinther et al. 2010 opted for case 2, but their record was from an older ice core which was measured using now outdated technology, with much lower resolution than the record we recovered recently. Thus, Vinther was dealing with a water isotope record that was already compromised in terms of preserved high-frequency variability, partly erased by the sampling procedure, and with high-frequency data further removed due to forward diffusion. With new technology, we have a record that preserves more high frequency information, thus we can use case 1 of back diffusion, which preserves the high-frequency data for climatic interpretations.

Another significant problem not stated by Vinther et al. 2010 is that diffusion models are highly uncertain. For example, both Johnsen et al. 2000 (in his seminal paper) and Jones et al. 2017 note that diffusion in excess of what models predict is evident in the GRIP and WAIS cores. At this time, it is accepted that firn diffusion models cannot always accurately reconstruct diffusion lengths, and should be approached carefully. This means that case 1 above is the preferred method, which estimates diffusion length directly from raw data, rather than relying on uncertain models. Since case 1 allowed for reliable Gaussian fits for the last 2.6 ka, we opted to use this method.

As Vinther et al. 2010 state, melt layers can occur at Renland. But, this is true of inland ice core records, as seen in 2012 on the Greenland ice divide (and known to have occurred in the past, e.g. in the GRIP record) as well as the WAIS Divide ice core in West Antarctica (Jones et al. 2017). Vinther et al. states that melt layers will cause "ringing" in the diffusion-corrected isotope data. Despite melt layers, neither GRIP nor WAIS exhibit ringing, and there is no evidence for 'ringing' from melt layers in our diffusion corrected interval to 2.6 ka for the annual signal, summer, and winter (see later comment for further discussion on melt in the RECAP core).

Even if there is still concern with the aforementioned methods in our paper, there is one important point that cannot be ignored: the pattern we observe in diffusion corrected data is also evident in the raw data. Thus, the diffusion correction we apply, and the associated summer and winter patterns, is not an artifact of the diffusion process.

We further prove that the seasonality of accumulation - not treated in the Vinther et al paper - cannot account for the pattern of summer, winter, and annual amplitudes. The seasonality of accumulation can weight diffusion toward whichever season has less snowfall. This is problematic because diffusion corrections assume a constant snowfall rate throughout the year. The high accumulation rate at Renland is beneficial in this regard, as any seasonality effect on diffusion is sufficiently damped to preserve the summer and winter patterns. Interestingly, a WAIS paper in prep. requires additional impurity information to constrain the annual signal since WAIS has much less accumulation than Renland.

When the strengths and weaknesses of the applied method are introduced it is meaningful to first

thereafter explore effects of climatic variability on the diffusion correction, as done with the CFM model, which is introduced to explore the effect of changes in accumulation seasonality. Please also discuss issues regarding the interplay between accumulation seasonality, insolation and sea ice changes. Can accumulation seasonality in reality considered to be constant or are the combined effects on diffusion correction larger?

The goals of our study are as follows: 1) Determine if the back diffusion method (rather than forward diffusion) was valid for the new high-resolution RECAP record. (We find that for the last 2.6 ka, back diffusion is possible with good Gaussian fits to the raw data). 2) Determine if seasonality of accumulation affected firn diffusion enough to alter the summer and winter signals. (We determine that the patterns in summer and winter on millennial timescales are robust). 3) Provide varying hypotheses for why these patterns might occur, leaving GCM and/or isotope-enabled modeling for future studies. There are some limitations to our methods. For goal 2) We cannot constrain variations in the seasonality of accumulation (i.e. we cannot claim this seasonality effect is a constant), thus centennial scale variability in summer and winter could be purely an artifact of firn diffusion, rather than a direct climate signal. For 3), we cannot conclusively determine a cause of the decreasing winter signal, for example. The modeling required to answer this question is outside the scope of our paper, but does provide opportunities for future papers to resolve this RECAP record, and hopefully other high-resolution records planned for the next decade.

To answer the reviewers question, we cannot directly treat the interplay between accumulation seasonality, insolation and sea ice changes without more advanced modeling, which will have to be reserved for future studies. Thus, we recognize the limitations of our study, and look forward to future results that validate the cause of Renland isotope seasonality. The seasonality effect cannot be considered constant, and we do not have sufficient impurity data or other proxies to constrain the seasonality effect beyond the work we have done using the CFM. It should be noted that the CFM results are an extreme case, meant to show that even a highly unlikely shift in seasonality cannot remotely come close to explaining the trends we see in the isotope data.

However, we will note that changes in seasonality of accumulation is the only climate variable that could likely introduce significant uncertainty to our diffusion correction results. While changes in temperature (driven by variability in insolation, sea ice, or other factors) can influence the diffusion length, these factors are included in our diffusion length estimation because it is calculated directly from the isotope data, and they would not introduce a seasonal bias. If, for example, we had estimated diffusion length by means of modeling using a thinning function and temperature estimate, climate variables could introduce uncertainty since they would be more difficult to constrain. L241 has been revised to clarify this:

"Climate variability such as changes in temperature and accumulation rate can influence the extent of diffusion that occurs in the firn column. While the method of estimating diffusion length directly from the water isotope record includes the effects of long-term changes in mean temperature and accumulation rate, changes in seasonality of accumulation creates uncertainty in the diffusion-correction calculation of the annual cycle."

Sea ice signals in the RECAP core: A change in sea ice does not always directly change into a similar change in d18O. See e.g. Holme et al., 2019, Faber et al. 2017, Sime et al. 2013, Divine et al., 2011. And for paleoclimate signals on Merz et al., 2015 and Li et al 2010. The authors are currently not demonstrating the processes in which a regional sea ice change near RECAP translates into a changed d18O signal. Existing literature is used to argue that a link is plausible through d18O, sea ice and AMOC, but the demonstration that this is actually the case for RECAP

is missing. Maffezzoli et al 2018 explored sea ice in the RECAP using impurities. The findings from this paper is extremely relevant to include here in order to argue for how sea ice variability is "seen" from the RECAP core using impurities. In the current approach the authors introduce a simple energy balance model to only because the variability in surface temperature (and therefore d180??) is not caused by insolation and thus indirectly argue that sea ice is the driver of the variability. This is not convincing. Effects on d18O and atmospheric circulation are not considered in this approach. I strongly suggest that the authors to include the use of (isotope) model simulations, moisture source tracking or similar to strengthen the argument on how the connection between sea ice and RECAP d18O variability must be used in order to demonstrate that the effects of Holocene sea ice variability is reflected in the d18O of RECAP.

As stated above, a limitation of our study is a lack of GCM and/or isotope-enabled modeling. We have made this more clear in the paper: This is already mentioned in L229, and we have added in section 3.2.3 and in conclusions that modeling would be beneficial as a future study. However, we do want to offer potential hypotheses for the pattern that we observe in summer and winter, and we have included identified text in the manuscript to alert the reader to any hypotheses we are making. We have reframed section 3.2.3 as a more general look at regional climate and sea surface conditions, and consider hypotheses for how these factors could influence the seasonal $\delta^{18}O$ record over the last 2.6 ka. We include a citation for Maffezzoli et al. 2019, which does demonstrate how the bromine enrichment proxy records sea ice in the RECAP core. However, the discussion in Maffezzoli et al. 2019 is limited to the last glacial period and deglaciation, so the impurities record cannot be used as a direct comparison to our results through 2.6 ka. As additionally mentioned in L309, Corella et al., 2019 does provide a record of impurities at RECAP in the Holocene. The following is the revised Section 3.2.3 "Regional climate variables":

"Indirect solar effects may influence other parts of the climate system, which then affect the local climate at Renland. A cooling trend in the North Atlantic is observed over the last 3 ka in records of glacial expansion, ice sheet growth, and increased drift ice (Miller et al., 2010), driven by a decrease in total annual insolation (Kaufman et al. 2009). In the RECAP core, a decrease in $\delta^{18}O$ is observed in winter, while the summer signal remains stable. While it is difficult to determine the cause of this trend without isotope-enabled modeling, we can hypothesize how possible mechanisms involving regional climate variability might influence seasonality of the isotope signal at Renland.

Since Renland is closer to the coast and open ocean than other inland ice cores, sea surface conditions could play a substantial role in Renland climatology (Holme et al., 2019). One possible factor is sea ice extent, which is both influenced by annual insolation (Muller et al. 2012) and influences total absorbed insolation at the surface due to albedo. A number of studies have documented an increase in sea ice cover in the North Atlantic and Fram Strait over the last 3 ka (Mller et al., 2012; Fisher et al., 2006; Jakobsson et al., 2010; Polyak et al., 2010). Sea ice extent has been previously studied through impurities in the RECAP core; iodine concentrations from the RECAP ice core suggest increasing sea ice over the last 3 ka (Corella et al., 2019; Saiz-Lopez et al., 2015), and bromine enrichment has been used to estimate sea ice conditions through 120 ka (Maffezzoli et al., 2019).

The formation of sea ice primarily occurs in winter, and increasing sea ice would be correlated with decreasing regional temperatures in winter and for portions of the shoulder seasons, depending on the timing of ice formation in fall and melt in late spring. A more open ocean regime at 2.6 ka, driven by higher total annual insolation, would keep winters warmer in coastal Greenland due to ocean heat contribution to the atmosphere (Screen and Simmonds, 2010). In recent centuries prior to the Industrial Revolution, lower total annual insolation and increased sea ice would dampen

the moderating effect the open ocean has on coastal winter temperatures, resulting in colder winters. Increasing sea ice is therefore consistent with increasingly colder winters at Renland, whereas summers would largely be immune to sea ice response since nearby water bodies have little to no summer sea ice. This may explain the similarity between the winter $\delta^{18}$O signal and total annual insolation at 71° N (Figs. 10c, 8c).

However, it is nearly certain that other regional climate variables have an influence on the $\delta^{18}$O signal in the RECAP core. At this time we lack a comparison to seasonality at other locations in Greenland, which would help to determine the extent to which the trends observed in the RECAP record are due to local or regional influences. The RECAP core is unique in that it has both high sampling resolution (0.5 cm, whereas most other cores are over 2 cm), and high accumulation rate (45 cm yr$^{-1}$ compared to 10–20 cm yr$^{-1}$ inland), allowing for a much more accurate diffusion-correction of the seasonal isotope signal. At Renland, we may also observe the effects of atmospheric and oceanic circulation patterns and sea ice extent, which can control the influence of local oceanic moisture in comparison to long-range transport. This would alter the $\delta^{18}$O signal through moisture source instead of a direct influence on local temperature (Johnsen et al., 2001; Klein and Welker, 2016). As we do not have records of isotope seasonality from inland Greenland, it is difficult to identify whether an effect such as this uniquely influences the RECAP isotope signal. These factors could be instead be further explored through additional modeling studies. "

RECAP and GRIP comparison (appendix A): This is interesting and important. The study argues that RECAP record signals of sea ice variability. Thus, it is important to demonstrate that RECAP is unique in this sense and that the same variability patterns are not found in the same cores. Differences in terms of accumulation and measurements resolution exists among the cores which creates issues, but the authors must present stronger arguments for why they find that RECAP variability is unique and driven by sea ice. This deserves to be treated thoroughly in the manuscript instead of the appendix.

After careful consideration, we have decided to remove the comparison to GRIP from the manuscript. Originally, we thought it was an interesting comparison, but the uncertainty on the GRIP record is too high for any meaningful comparison and it will only introduce confusion when interpreting the RECAP core as a unique coastal ice core record. Here, we outline the reasoning for the uncertainty on the GRIP record, and how we have modified our conclusions on the RECAP record after removing this comparison.

First, the GRIP isotope record has much lower sampling resolution (2.5 cm) than the RECAP record (0.5 cm). This is because the GRIP record was measured using discrete ice samples, prior to the advancement in technology that allowed for the continuous flow method used in analysis of the RECAP core. Additionally, GRIP has a much lower accumulation rate (23 cm yr$^{-1}$ compared to 45 cm yr$^{-1}$ at Renland), resulting in lower resolution sampling with time.

Because GRIP has a much lower sample resolution, this introduces significant uncertainty to the analysis and diffusion correction. The lower accumulation rate also makes the diffusion correction more susceptible to seasonal bias in accumulation. We could rule out this issue at Renland by using the CFM test, but preliminary CFM models of WAIS Divide (which has similar temperature and accumulation rate to GRIP) have indicated a much stronger influence on the isotope signal. For WAIS Divide, additional proxy records such as chemistry data can be used to help constrain any changes in seasonality of accumulation (manuscript by Jones et al., will be submitted soon), but we do not have the needed additional data for GRIP. Therefore, we cannot rule out the possibility that any trends observed at GRIP are not simply due to changes in seasonality of accumulation,

making a comparison to RECAP less meaningful.

For example, we have calculated the mean rate of change over the period from 2.6 ka–present for the normalized amplitude. While the rate of change is larger at RECAP (0.075 ka$^{-1}$) than at GRIP (0.034 ka$^{-1}$), the uncertainty on the amplitude for GRIP effectively overwhelms the signal. This is demonstrated by the gray shading in the figure shown here, which is substantially larger for GRIP than for RECAP.

Because we do not have an inland ice core with high enough sampling resolution and accumulation to accurately reconstruct the seasonal signal, we cannot claim that the trends observed in the RECAP isotope signal are unique. However, the RECAP record is unique in that it is the only existing Greenland core which we are able to use for this type of analysis, making it a very valuable

[Figure]

FIGURE 1. Comparison of RECAP and GRIP seasonal data. (a) The summer and winter $\delta^{18}$O signal for both RECAP (red and blue) and GRIP (orange and green). (b) The annual amplitude for RECAP (black) and GRIP (purple) shown as both $\delta^{18}$O (right axis) and normalized to the maximum $\delta^{18}$O for each record (left axis).

record. We have revised part of Section 3.2.3 and the Conclusions to reflect this:

Section 3.2.3: "However, it is nearly certain that other regional climate variables have an influence on the $\delta^{18}O$ signal in the RECAP core. At this time we lack a comparison to seasonality at other locations in Greenland, which would help to determine the extent to which the trends observed in the RECAP record are due to local or regional influences. The RECAP core is unique in that it has both high sampling resolution (0.5 cm, whereas most other cores are over 2 cm), and high accumulation rate (45 cm yr$^{-1}$ compared to 10–20 cm yr$^{-1}$ inland), allowing for a much more accurate diffusion-correction of the seasonal isotope signal. At Renland, we may also observe the effects of atmospheric and oceanic circulation patterns and sea ice extent, which can control the influence of local oceanic moisture in comparison to long-range transport. This would alter the $\delta^{18}O$ signal through moisture source instead of a direct influence on local temperature (Johnsen et al, 2001; Klein and Welker, 2016). As we do not have records of isotope seasonality from inland Greenland, it is difficult to identify whether an effect such as this uniquely influences the RECAP isotope signal. These factors could be instead be further explored through additional modeling studies."

Conclusions: "At this time, Renland is the only available Greenland ice core that has high-resolution sampling and high accumulation rates. These factors are necessary to rule out seasonality of accumulation effects on diffusion, allowing for an interpretation of the summer and winter patterns for the last 2.6 ka. Whether Renland is unique in its downward trend in winter values remains to be seen."

Melt: The authors address the effect of melt in L264ff. For the Holocene, frequent summer melt must be expected at RECAP. It is unclear how melt is influenced the d180 seasonality trough vertical mixing of summer and winter layers. Given the importance of reconstructing a correct summer/winter d18O signal for the conclusions of this paper, the role of melt on seasonality is relevant to address further. The authors are free to find the best approach to address this challenging issue.

Vinther et al. 2010 state that the shape of the diffusion-corrected water isotope data will be altered by melt layers, resulting in 'ringing' effects of spurious high-frequency oscillations. In the last 2.6 ka - the interval of time for which we interpret the annual signal - we do not observe ringing in diffusion-corrected data. This strongly suggests that melt layers have had a minimal impact. As mentioned by Johnsen et al. 2000, diffusion in a firn column without melt layers is expected to produce isotope data in which the fit to the diffused portion of the PSD is a Gaussian (Johnsen et al., 2000). Indeed, this expectation holds for the last 2.6 ka in the RECAP isotope record.

This section has been expanded to better clarify the potential effects of melt on the seasonal layers, including a better description of the vertical mixing of seasonal layers, discussion of the 'ringing' effect in the spectrum, and further comparison between data in Taranczewski et al. (2019) and our seasonal data:

"Since the Renland site is subject to warm summer temperatures, we must also consider the possible effects of melt layers on the seasonal signal. A summer melt event could cause surface snow with a relatively high $\delta^{18}O$ signal (ie. near the seasonal peak) to percolate vertically through the firn column, mixing with the underlying winter layer which has a lower $\delta^{18}O$ value. This mixing would cause the preserved $\delta^{18}O$ value of the winter layer to increase, resulting in a decrease in the annual amplitude. Alternatively, melt water which refreezes in the firn as an ice lens can produce a local barrier to further diffusion. In either case, it is important to consider the extent to which

melt layers could influence the recorded water isotope signal, which we do so by examining both the isotope data and the melt layer density in the RECAP core.

Diffusion in a firn column without melt layers is expected to produce isotope data in which the fit to the diffused portion of the PSD is a Gaussian (Johnsen et al., 2000). Substantial alteration of firn processes due to melt, either through liquid water mixing or an ice lens barrier to diffusion, would likely influence the shape of the spectrum. Over the time period in which we reconstruct the annual signal, we do not observe degradation of the Gaussian fit in the 300-year windows of PSD, indicating that melt has not significantly influenced the isotope data. Additionally, it has been noted that melt layers can cause a 'ringing' effect in the diffusion-corrected data, resulting in spurious high-frequency oscillations (Vinther et al., 2010). We do not observe this effect in the diffusion-corrected isotope data at RECAP over the last 2.6 ka.

The density of melt layers at RECAP was measured by Taranczewski et al., (2019), determining a high-resolution record for the last 2.1 ka. The ratio of snow water equivalent of a melt layer to the respective annual layer is characterized as the annual melt ratio (AMR), which over the last 2.1 ka has an average value of less than 2% and is therefore a very small fraction of the total annual ice volume (<1 cm for 45 cm ice per year). The AMR exhibited some centennial variability with a few distinct periods of increased melt, but did not demonstrate a long-term trend over the last 2.1 ka (Taranczewski et al., 2019). Furthermore, during brief periods in which there is a $\sim$1% increase in AMR (ie. 1850–1700 yr b2k, 200 yr b2k–present), increases in melt layer occurrence would likely serve to decrease the annual amplitude, which we do not observe. Based on this evidence, it is likely that the presence of melt layers is not significantly influencing the seasonality trends observed in the $\delta^{18}$O signal. "

Sec 3 "Results and discussion". I encourage the authors to separate the results from discussion to not mix up results from this study with hypotheses entirely based on other studies.

We find that the flow of the paper is better with a combined 'Results and discussion' section, so as to not repeatedly jump between analysis of interannual climate variability and seasonality. However, we do agree that results and hypotheses should be more clearly separated, and as such we have added statements to make it more clear when we are transitioning from discussing results to hypotheses. For example, sections 3.2.1–3.2.2 are primarily results directly derived from our data (with the exception of part of the discussion of melt layers, in which the data that comes from Taranczewski et al., 2019 is clearly stated), and section 3.2.3 is primarily a discussion based on other studies and our hypotheses. To clarify this, we have added the following statement near the beginning of section 3.2.3:

"While it is difficult to determine the cause of this trend without isotope-enabled modeling, we can hypothesize how possible mechanisms involving regional climate variability might influence seasonality of the isotope signal at Renland."

In section 3.1, we clarify the transition from results to discussion with the following statement: "...considering the common geographic region, we can hypothesize how potential mechanisms could link the two records."

Comparison with VM28-18: It is relevant and meaningful to compare the data to this core and the authors argues for this in a good way. It is however symptomatic for this analysis that this hypothesis is entirely driven by other studies and the data and analysis in this study does not convincingly support this hypothesis (full span of the records r=0.34). In the current format of the

paper this link with the Bond cycles provide an argument for connecting d18O records to sea ice. I suggest the authors to reconsider whether this approach is optimal. If yes, then the analysis of the correlation between d18O and Bond cycles must be statistically stronger to ensure that this correlation is not just noise and coincidence. If no, then please explore d18O and sea ice in alternative ways as explained later in this review.

We agree that there is not enough evidence to draw any conclusions about a correlation between our records and the VM28-14. As stated, it could arise purely by chance due to noise in the system, and the correlation is much too low for significance. While it is possible that a better timescale for VM28-14 could result in an increased correlation (as discussed in a later comment), we cannot completely discount the possibility that this relationship is just coincidence. Yet, discussion of Bond Cycle in both the Holocene and glacial has persisted for decades, and this is yet another breadcrumb in that debate. There is a chance that the decadal variability at Renland is related to these Bond Cycles, and we provide Figure 8 as evidence of a potential link. In this paper, we cannot make a definitive conclusion about any such relation, and we are very cautious in how we present the relationship (ie. L192 "...we cannot rule out the possibility that this similarity arises due to random noise in the climate system..."; L226 "While the correlation ... is not conclusive evidence of this relationship, there is a plausible physical connection..."). However, the information is important for future modeling that could explore links between ocean circulation and decadal variability expressed through the hydrologic cycle at Renland.

Sec 2.5 Community Firn Model (CFM): This section has several issues. The motivation for introducing the CFM, is only mentioned in the end of the section and is unclear for the reader. Please expand the text to explain accordingly.

We have moved the description of the purpose of the CFM test from L241 (results section) to L162 (methods sections, where CFM is introduced):

"Climate variability such as changes in temperature and accumulation rate can influence the extent of diffusion that occurs in the firn column. While the method of estimating diffusion length directly from the water isotope record includes the effects of long-term changes in mean temperature and accumulation rate, changes in seasonality of accumulation creates uncertainty in the diffusion-correction calculation of the annual cycle. If there is a seasonal accumulation bias (i.e. more snow in summer than winter), the drier season will be subject to greater isotopic attenuation due to firn diffusion. Because we cannot selectively diffusion-correct the seasonal isotope signal for accumulation bias, we must assume constant seasonality of accumulation. Therefore, the drier season will be under-corrected for diffusion, and the wetter season will be over-corrected to a lesser extent, resulting in potential inaccuracies in the amplitude of the seasonal signal. While there is no current method for reconstructing past seasonality of accumulation, we can utilize a series of tests to determine the extent to which the diffusion-correction calculation for $\delta^{18}$O could be affected by seasonally-biased accumulation.

We use the Community Firn Model (CFM) (Stevens et al., 2020) to test the effects of seasonally-biased accumulation on firn diffusion in the ice core. We test five 490-year scenarios for isotope evolution..."

Using CFM with input from MAR forced by ERA-Interim creates a long chain of uncertainties and known model biases. Especially for the Renland Ice Cap which is not represented well in most model grids. Nearby coastal and AWS weather stations exists, please demonstrate that the model results are in line with nearby observations. The outcome/results of this method are not clear from

the text (but clearer when fig 10 is shown)

Nearby weather stations have only been recently installed and do not have as long records (ie. the closest PROMICE station was installed in 2008), and coastal stations with longer records are at significantly lower elevation and different latitudes. The output using the ERA forcing data is validated using existing weather stations and satellite data (Fettweis et al. 2017), and is expected to be the best available estimate of local climate.

Furthermore, a key point here is that our analysis with the CFM uses the MAR data only as an estimate of modern-day accumulation, in order to select a highly unlikely end-member possibility of changing accumulation seasonality; therefore the analysis is not dependent on getting present-day climatology exactly correct. Any biases in the MAR data would not substantially change the modeled isotope values as we are already using extreme scenarios, and existing climate records do not indicate a change in seasonality of accumulation as large as what we model. To clarify this in the text, we have added the following statement to L261:

"While this analysis may include uncertainties in the MAR reanalysis data, it provides an end-member possibility for highly unlikely shifts in seasonality of accumulation, demonstrating that the effect on $\delta^{18}$O seasonality is minimal in comparison to observed trends."

Methods: Please add a short separate section for information regarding the location and drilling of the RECAP core (see similar in Holme et al, 2019 and Maffezzoli et al 2018)

The following text has been added to the beginning of the Methods section:

"The RECAP core was drilled near the summit of the Renland ice cap (-26.75, 71.2333) from May-June 2015. The drilling location is approximately 2 km away from the location of a core previously drilled in 1988 (Johnsen et al, 1992). The 584 m RECAP core, drilled to bedrock, contains a continuous record extending through 120 ka."

Naming RECAP vs Renland: The manuscript refers to the ice core as Renland (except for the Figure caption of fig 2) The Renland core was drilled in 1988 and RECAP (REnland ice CAP) ice core in 2015 nearby, both on the Renland Ice Cap. I suggest that the text is corrected in a suitable way to avoid misunderstandings and reflect the correct naming of the core.

This is clarified throughout the paper, such that "Renland" refers to the peninsula and its local climate, and "RECAP" refers to the ice core and water isotope record.

Comparison with existing record (Renland 1988): How does the former Renland core compare to the newer RECAP core? Do we see the same signal for the Holocene period or are there any deviations? Other than differences in instruments and precision the cores are not expected to be very different, but it would be good to demonstrate the agreement with the previous core. This can for instance be done in one of the very first figures.

The former Renland 1988 record was measured at much lower resolution and potentially has dating inconsistencies, resulting in a vastly different isotope signal. For example, the 8.2 ka event, clearly visible in the new core, is not resolved in the 1988 record. Lack of tie points likely contributed to dating issues in the 1988 record, making a direct comparison between the two cores difficult. For this reason, we have chosen not to include a comparison as it may only introduce confusion. However we have added citations for prior publications including the Renland 1988 data (Johnsen et al.

1992) to the beginning of the Methods section, if the reader wishes to make their own comparison.

Figures: The manuscript is relatively long and have several figures of less relevance. It would strengthen the quality and readability of the manuscript if some of these less relevant figures would be placed in the supplementary instead. Examples are Fig 5, Fig 6,7

Figure 5 has been added to the supplementary material, as a panel of Figure B3, and the caption for Figure B3 has been revised accordingly. However, we maintain that Figs 6 and 7 are relevant enough to remain in the main text of the paper. Fig. 6 and 7 both show the effect of diffusion on higher frequency bands, supporting our decision to focus on analysis of 0–8 ka bp, and 0–2.6 ka bp for the annual signal. Fig. 7 shows that the diffusion correction does not influence the shape of the frequency band relative amplitudes, which is important to consider when comparing the 15-20 year band to core VM 28-14.

**Minor comments**

Figure 1: Please add the description of the location markers and the reasons for the different colors (ice, marine etc) to the figure caption

The caption has been revised:

"A map of the study region shows the RECAP drill site (red) is located on the east coast of Greenland. The Renland ice cap is approximately 80 km wide, and is isolated from the Greenland ice sheet. The drill site is near the summit of the ice cap, at a location of -26.75, 71.2333. The locations of several other Greenland ice cores are also shown for reference (blue), as well as North Atlantic sediment core VM 28-14 (green) (Pawlowicz, 2020)."

L29: This reference "Noone and Simmonds 2004" concerns conditions in Antarctica and is not suitable here. Please use a better reference, e.g. evidence from other analysis on the same core or similar (e.g. Holme 2018)

The reference to Noone and Simmonds 2004 has been changed to Holme et al. 2019.

L29-30: "These climate parameters are recorded in the ice core water isotope (i.e. $\delta$ 18 30 O) record through changes in condensation temperature at the time of precipitation". This statement does not agree with the last 10-15 years of isotope and ice core research. Please add a few lines with supporting newer references that explain how the ice core isotopes is an integrated signal of several processes from source to site.

This statement has been revised:

"Polar ice core water isotope records are correlated to condensation temperature at the time of precipitation (Dansgaard, 1964; Dansgaard et al., 1973; Craig and Gordon, 1965; Merlivat and Jouzel, 1979; Jouzel and Merlivat, 1984; Jouzel et al., 1997), and integrate across regional ocean and atmospheric circulation patterns and sea surface conditions along the moisture transport pathway (Johnsen et al., 2001; Holme et al., 2019)."

L60-65: This should be well known to most readers, and is not relevant in the method section

This section has been shortened and instead amended to L69:

"...This technique produces $\delta^{18}O$, $\delta D$, and $\delta^{17}O$ water isotopes, where delta notation refers to a ratio of heavy to light isotopes measured with respect to a standard (Vienna Standard Mean Ocean Water) and is expressed in parts per thousand (per mille or ‰) (Dansgaard, 1964). Water isotope data has sub-mm nominal resolution..."

L88-98: This text is mainly "textbook material" and does not belong in a method section. Please shorten this.

LL88-96 has been kept as a brief explanation of diffusion, but is shortened:

"In the firn column, vapor diffuses along concentration and temperature gradients. Exchange of water molecules takes place between unconsolidated snow grains and vapor, attenuating the seasonal water isotope signal and acting as a smoothing function (Whillans and Grootes, 1985; Cuffey and Steig, 1998; Johnsen et al., 2000; Jones et al., 2017a). Solid-phase water isotope diffusion in ice below the firn column occurs at a much slower rate, with diffusivities increasing for warmer ice near bedrock. Over thousands of years, solid-phase diffusion can have a substantial impact on the attenuation and smoothing of high-frequency signals (Itagaki, 1967; Robin, 1983; Johnsen et al., 2000; Gkinis et al., 2014; Jones et al., 2017a)."

L164. Please argue for the choice and robustness of assuming a 4 permill sine wave for the amplitude which is larger than shown later.

The 4‰ sine wave is chosen based on the mean annual signal in the upper 10 years of the core, for which we can assume there is little to no effect from diffusion. Because it is not possible to diffusion-correct the firn column for the last 76 years, this is the closest estimate we can make for the MAR period 1958-1978. LL164-165 has been revised to clarify:

"A constant amplitude (4‰) sine wave is used to represent the annual isotopic variability (Fig. B4c), based on the mean amplitude of the relatively un-diffused most recent 10 years of the $\delta^{18}O$ signal."

L163: The authors chose to force the MAR model with re-analysis data in a time period pre-satelite era. Biases in this time period is expected to be large in the Artic region given the very limited data. Please discuss, maybe based on Fettweiss et al 2017, whether this choice is expected to introduce new biases to the results.

The MAR model is validated with several other datasets, including ice core records and surface mass balance measurements which span the period used here. It is also validated with satellite data starting in 1979, and PROMICE data weather station data starting in 2007, improving model output for earlier time periods (Fettweis et al., 2017).

We do not expect that this choice would introduce biases in the results. In our model, we show that even with large bias and variability in seasonality of accumulation, the diffusion correction of the water isotope signal is not significantly influenced. The input data would have to be substantially different to change this result, and we believe this is unlikely. See previous comment for changes made to text to clarify this.

L165: exchange the word predict with simulate. (The model simulates temperatures in the past)

Done.

L165: "...July-September receive the most precipitation on average". How much more precipitation comes during summer than winter? Please add the relevant numbers to support this statement including relevant statistics as precipitation is highly variable on Greenland, especially on the coast.

This statement is supported by Fig. 5 and Fig. B3, which have been combined in the appendix. Seasonal accumulation bias is shown in [previously] Fig. 5, and standard deviation has been added to show variability. A reference to the figure has been added to the text for clarity.

L167-172. Please be clear how and if the sum of annual accumulation varies from year to year in each of the scenarios.

The sum of annual accumulation is the same for all scenarios, with just the monthly distribution changing; L166–167 is revised to clarify:

"...The mean annual accumulation from the 20-year MAR period is used for all model scenarios, with the following variations on the seasonality of accumulation applied..."

Fig 5,6,7: I suggest that these figures are added to the supplementary material instead.

Figure 5 has been added to the supplementary material, as a panel of Figure B3, and the caption for Figure B3 has been revised accordingly. However, we maintain that Figs 6 and 7 are relevant enough to remain in the main text of the paper. Fig. 6 and 7 both show the effect of diffusion on higher frequency bands, supporting our decision to focus on analysis of 0–8 ka bp, and 0–2.6 ka bp for the annual signal. Fig. 7 shows that the diffusion correction does not influence the shape of the frequency band relative amplitudes, which is important to consider when comparing the 15-20 year band to core VM 28-14.

Figure 5: Please add the standard deviations to the figure. The figure B3 demonstrate large variability.

The standard deviations have been added to accumulation and temperature values.

L191: Could the variability in correlation strength be due to methodology/age model? Are there reasons to believe that age model differences in the Bond core can explain this mismatch? Please discuss this.

L192 has been expanded to introduce this possibility:

"...that they are statistically significant. Additionally, it is possible that inaccuracies in the dating model for VM28-14 (on the order of ± 200–500 years) (Bond et al., 2001) could account for some of the reduced strength in correlation outside of the period from 2.5–6.4 ka. While we cannot rule out the possibility that this similarity..."

L192: Please clarify that the p-values are calculated on time series without significant autocorrelation

L192 has been revised:

"Both relationships have a p-value <0.01, indicating that they are statistically significant, and the time series are not significantly autocorrelated."

L214: If this is a finding from Holme et al 2019, please connect this statement with a repeated reference to this paper.

The repeated reference to Holme et al. 2019 has been added.

Fig 9: is panel (a) necessary to plot? Removing this would make this figure a little less chaotic.

We believe that panel (a) is useful to show the comparison between raw and diffusion-corrected data for the full annual record. However, it is not critical to this figure, so it has been moved to the appendix to make Figure 9 easier to interpret.

L250:"The model results show that the Renland diffusion correction is minimally influenced by seasonality of accumulation (Fig. 10)". How is this clear? Please argue with numbers and a changed design of fig 10 (see below)

Fig. 10 has been revised (see below) and LL255-256 has been revised to clarify this:

"Seasonally-biased accumulation scenarios can be compared to the constant accumulation scenario, in which each month receives the same amount of accumulation and the diffusion-corrected amplitude matches the pre-diffusion signal. In comparison to the constant accumulation scenario, there is a maximum 15.7% decrease in the annual amplitude of diffusion-corrected isotope values for varied accumulation scenarios, which is a direct result of the bias in the diffusion-correction. Thus, in the unlikely case that accumulation shifted from a constant scenario to a seasonal-bias over the last 2.6 ka at Renland, we could expect up to a 15.7% offset from the true value of the annual amplitude... Furthermore, the annual amplitude in the observed RECAP water isotope record increases by approximately 50%...."

Fig 10: This plot is difficult to read and interpret. Either plot these on two different plots for summer and winter, or consider plotting the anomaly from the mean instead. In the current version of the figure it is unclear what the authors wants to demonstrate.

This figure has been updated to have three panels: 1) summer; 2) winter; 3) annual amplitude:

[Figure]

Figure A1: Please improve the figure caption to separately describe the left and right panel

This figure has been removed from the paper; see previous comment about the RECAP/GRIP comparison for details.

Appendix A1: It is argued that the effect of sea ice on GRIP is muted, but Fig A1 right panel show that both experience an increase in amplitude, but GRIP looks muted as the range of d18O is larger for GRIP. Please plot these on comparable scales. And argue sufficiently that RECAP is unique compared to GRIP. The conclusions about the comparisons in this section are currently not fully supported by values from analysis of the used models and data. E.g. L342: "A caveat in analyzing this data is that accumulation may have a greater seasonal bias at GRIP". Please support all statements in the appendix with values e.g. from MAR, CFM and ice cores.

This figure has been removed from the paper; see previous comment about the RECAP/GRIP comparison for details.

L335: Following open access style please provide all data types (raw and formatted) online before publication.

The data is currently in review for availability on Pangea (availability statement has been updated to reflect location).

---

## Author Comment (AC2) · 20 May 2020

**Response to Referee #2**

Hughes et al. present a new oxygen isotope temperature proxy data set covering the Holocene from an ice core drilled on the Renland Peninsula ice cap in eastern coastal Greenland. The isotope record is obtained on a high nominal resolution from Continuous Flow Analysis coupled to Cavity Ring-Down Spectrometers. Originating from this high-resolution data set, the authors use spectral analyses to estimate diffusion lengths and to investigate the extent to which the isotope (climate) signal of certain frequencies and frequency bands survives the diffusional smoothing throughout the Holocene. By means of deconvolution, they reconstruct the seasonal summer and winter isotopic time series. Subsequently, the authors discuss the relationship of the isotope records' decadal variability with regional sea ice and ocean circulation changes as well as potential causes of the observed decrease in winter isotope values over the last 2.6 ka.

The paper presents a new proxy data set at high resolution and discusses its relevance to assess regional climate information. This is an important contribution since the interpretation of isotopic data from coastal ice cores is an up-to-date topic and because the investigated site has only been studied previously at relatively low temporal resolution. The paper thus fits well within the scope of Climate of the Past. I would like to congratulate the authors on their elaborate analysis of the isotope data; the methods are sound, the paper is well written and structured and the figures overall adequately present and support the findings of the paper. However, my major concern with this work pertains to the interpretation of the results in terms of "high-frequency climate oscillations" and regional oceanic conditions (sea ice, ocean circulation). In the current state, I see only relatively weak evidence in the presented data supporting the claims made in title and abstract. Thus, I would recommend publication of the paper after a major revision of these general issues, which I outline below. In addition, I give a list of specific as well as technical comments you might want to address.

We thank the reviewer for taking the time to provide a thorough analysis with constructive criticism. Below we individually address each comment, outlining changes we have made to improve the manuscript. All line numbers are in reference to the original manuscript.

**General comments**

A first major point is that the main conclusions the authors give split up in two seemingly separate issues, i.e. the relationship of the decadal variability with sea ice conditions and/or ocean circulation, and the explanation of a decreasing winter trend by increasing sea ice. These two interpretations are presented rather disconnected throughout the manuscript, partly due to the combined structure of results and discussion, but also in the final overall conclusions section. As a result, the reader is somewhat left puzzling what the overall essence of this paper is, how the two presented main results connect with each other and what we in general learn about interpreting isotope records from coastal regions.

While we see the value in having a separate results and discussion section, we believe that this would likely make the structure of the paper more disconnected and difficult to follow - jumping between results and discussion of high-frequency variability over 8 ka and variability of seasonality over 2.6 ka may be more confusing and flow poorly. However, in an effort to better link the two sections, we have revised the conclusions section to first reintroduce the core and record, then briefly discuss conclusions from each section, and tying them together at the end:

"The RECAP ice core from coastal East-Central Greenland contains a high-resolution water isotope record of the Holocene, obtained using continuous flow analysis (Jones et al., 2017b). The coastal proximity of the Renland ice cap makes the isotope record subject to influence from sea surface conditions (Holme et al., 2019). The record preserves annual variability for the last 2.6 ka, interannual variability for the last 5.5 ka, and decadal variability for the last 8 ka. We perform a diffusion correction calculation on the annual signal, based upon the diffusion length fitting routine of (Jones et al., 2017a).

The diffusion correction calculation for annual variability requires testing by a firn model to determine if biases in seasonality of accumulation can alter the patterns observed in summer and winter extrema. Since diffusion corrections assume constant annual accumulation rates, diffusion correction can over- or under-estimate the summer and winter signals. We utilize the Community Firn Model (CFM) (Stevens et al., 2020) to test an extreme case of shifts in seasonality of accumulation, from constant accumulation to summer-fall weighted accumulation, as given by MAR (Fettweis et al., 2017). We cannot rule out that centennial variations in summer and winter isotope values could be the result of seasonality of accumulation and the associated diffusional effects. However, the millennial-scale trend is robust: The summer extrema are relatively steady, and the winter extrema are steadily declining with a greater magnitude of change than can be accounted for by changes in seasonality of accumulation. We suggest that increasing sea ice driven by decreasing total annual insolation over the last 3 ka could cause the winter decline; however, other regional climate variables such as atmospheric and oceanic circulation may also exert an influence on the seasonal isotope signal. Future isotope-enabled modeling studies are needed to constrain how these factors are reflected in the RECAP $\delta^{18}$O signal..

The interannual and decadal signals are variable at millennial and centennial timescales, with a somewhat constant trend over the last 8 ka. This is to be expected, as the background climate state of the Holocene is also stable. We note that variability centered on 20 years has millennial-scale variations that appear similar to Bond Events recorded in a North Atlantic sediment core off the west coast of Iceland (VM 28-14) (Bond et al, 2001). The relationship is significant, but with low correlation for the full 8 ka record. However, VM 28-14 has large dating uncertainty, which could lessen the correlation. This potential relationship may be of interest for future isotope-enabled modeling, as this sort of millennial variability is pervasive in the North Atlantic (Bond et al, 2001).

Future ice core studies at locations with the annual signal preserved will provide additional constraints on spatial and temporal variability in the annual water isotope signal. At this time, Renland is the only available Greenland ice core that has high-resolution sampling and high accumulation rates. These factors are necessary to rule out seasonality of accumulation effects on diffusion, allowing for an interpretation of the summer and winter patterns for the last 2.6 ka. Whether Renland is unique in its downward trend in winter values remains to be seen. We suggest careful planning for future studies of the annual signal: 1) If site accumulation rate is low, impurity data will be needed to constrain the seasonality of accumulation; and 2) Isotope-enabled modeling should be used once new records are obtained, which will improve our understanding of regional climate dynamics."

Additionally, we do agree that results and hypotheses should be more clearly separated, and as such we have added statements to make it more clear when we are transitioning from discussing results to hypotheses. For example, sections 3.2.1–3.2.2 are primarily results directly derived from our data (with the exception of part of the discussion of melt layers, in which the data that comes from Taranczewski et al., 2019 is clearly stated), and section 3.2.3 is primarily a discussion based on other studies and our hypotheses. To clarify this, we have added the following statement near the beginning of section 3.2.3:

"While it is difficult to determine the cause of this trend without isotope-enabled modeling, we can hypothesize how possible mechanisms involving regional climate variability might influence seasonality of the isotope signal at Renland."

In section 3.1, we clarify the transition from results to discussion with the following statement: "...considering the common geographic region, we can hypothesize how potential mechanisms could link the two records."

This problem is also reinforced by the presented evidence to support the interpretation, which is rather weak in my opinion.

The title of the paper alludes to observed climate oscillations with a high frequency. Firstly, I would not denote decadal variations as being high frequency, but more importantly, the authors present a rather simplistic and too deterministic view on climate variability. Strictly speaking, an oscillation is defined as a periodic variation in time with a fixed frequency, or in a coupled system, a superposition of several frequency modes. For climate variability, this deterministic view is not necessarily the case. The Atlantic Multidecadal Oscillation might indeed appear to be a quasioscillatory phenomenen, but very likely it does not exhibit a strict 20-year cyclicity, as suggested in the paper (P3 L45). By contrast, the spectrum of the NAO is essentially white, so these are rather random variations than "subdecadal climate [or] multi-year cycles" (P3 L46-48). This also applies to the "Bond cycles". There seems to be no single deterministic cause (Wanner and Btikofer, 2008), the 1500-yr periodicity is likely an artifact (Obrochta et al., 2012), and the variations could be simply compatible with internal climate background variability (see also the discussion of the statistics of D-O events as their glacial counterparts; e.g. Lohmann and Ditlevsen (2018)). I would suggest that the authors critically revise the text passages where these phenomena are discussed; especially focussing on toning down statements of Bond events having a 1500-yr periodicity (P2 L35, P11 L208, P17 L318) an rewriting the respective introductory paragraph (P2 L35 – P3 L42). Here, maybe use AMOC as a starting point and then develop its influence on local climate and sea ice?

To address this, we have changed phrases such as 'oscillation' and 'cycle' to more general 'variability' or 'signal', where appropriate. (ie. L9, L36, L45, L46, L47, L48). We have also changed phrasing regarding Bond events from '1500-year cycle' to 'millenial-scale variability', and changed the title of the paper to "High-frequency climate variability in the Holocene from a coastal-dome ice core in East-Central Greenland". We have also revised the introductory paragraphs LL35–42 (and LL 43–51) as suggested:

"The modern instrumental record documents a number of influences on climate variability in the North Atlantic and Arctic, which may exert an influence on the local climate at Renland. The Atlantic Meridional Overturning Circulation (AMOC) controls heat transport to the Arctic and can have a substantial effect on Arctic climate over long timescales. Heat is supplied to the Arctic via the Norwegian Current, carrying warm Atlantic water to the Arctic (Polyakov et al., 2004); changes in heat supply and northward advection will influence atmosphere-ocean heat exchange, regional Arctic climate, and sea ice cover (Muilwijk et al., 2018). Heat distribution through AMOC controls sea surface temperature and drives the Atlantic Multidecadal Oscillation (AMO), which is observed in prior Greenland ice cores and influences sea ice cover in the Arctic (Chylek et al., 2011)with approximately 20-year variability. Subdecadal climate signals are also observed in the Arctic, such as the North Atlantic Oscillation (NAO), which is currently expressed as shifting sea-level pressure differences between the Subtropical High and Subpolar Low. This multi-year variability influences

temperature, precipitation, sea ice distribution, and ocean circulation across the entire North At-
lantic (Hurrell et al., 2009), and it is recorded in multiple proxy records including tree-ring data
and central and western Greenland ice cores (Barlow1993, Appenzeller1998).

On longer timescales, changes in sea surface conditions are also reflected in North Atlantic sediment
cores. Sediment cores from the North Atlantic show that millennial-scale climate variability oc-
curred throughout the Holocene (Bond et al., 1997, Bond et al., 2001), referred to as Bond events.
The mechanism forcing this variability is still under debate, but it is potentially driven by solar
forcing (Bond et al., 2001) or internal climate dynamics such as interactions between the ocean and
atmosphere (Wanner et al., 2015). The effects are most prominent in the Arctic, likely transmitted
to lower latitudes through AMOC (Bond et al., 1997, Bond et al., 2001, DeMonocal et al., 2000)."

The comparison of the 15–20 year variability with the Bond curve (Figs. 7 and 8) is also crit-
ical in this regard. While the title of the paper suggests that you observe oscillations in the isotopic
time series, what the authors actually show in the figures here is the time evolution of the isotopic
variability (if I understand correctly, either the average spectral power or the variance; see specific
comments) in the frequency bands. While this is certainly a very interesting analysis, the difference
to "normal" variations in the time series should be made clearer to the reader in the text and the
title of the paper.

Our method of spectral analysis provides information about the amplitude of a signal at a given
frequency for each 300-year PSD window, which is different from an analysis of variance. We have
clarified this issue by expanding the methods section, L140–143 which is revised to the following:

"Further analysis of the PSD can be used to determine the amplitude of climate signals on an
interannual to decadal scale (Jones et al., 2018). While the raw isotope data is comprised of a
continuum of climate variability at all frequencies, it is possible to separate out the strength (ie.
the amplitude) of varying frequency bands within that continuum. The power ($P_i$) of individual
frequencies from 1–20 years are identified for each 300-year PSD window, as well as the frequency
bands of 3–7, 7–15, 15–20, and 20–30 years. For frequency bands, the average power is calculated
by integrating across the power spectrum within each frequency band:

$$(1) \qquad P_i = \frac{\int_{f_a}^{f_b} P(f)\mathrm{d}f}{f_b - f_a}$$

where $f_a$ and $f_b$ represent the lower and upper frequency limits, respectively. Because the fre-
quencies are not evenly spaced, this method ensures that the average is not biased towards higher
frequencies. The amplitude (ie. the strength) is calculated as the square root of $P_i$. In our analysis,
the strengths of individual frequencies are normalized to the strength of the annual signal in the
most recent window, and the strength of each frequency band is normalized to its most recent value.
This produces a time series of the relative strength of isotopic variability for several individual fre-
quencies (1, 3, 5, 7, 10, 20 years) and interannual to decadal frequency bands (3–7, 7–15, 15–20,
and 20–30 years) (Figs. 6, 7)."

Additionally, we have added to the appendix a figure comparing the strength of decadal vari-
ability to the raw isotope data, demonstrating the difference between variations in raw data and in
spectral analysis. We have also changed the title of the paper to "High-frequency climate variability
in the Holocene from a coastal-dome ice core in East-Central Greenland"

Moreover, a discussion of a possible mechanism would be welcome: what could link temporal changes in drift ice to changes in isotopic variability, i.e. non-stationarity, within the 15–20 year frequency band? Alternatively, what is the significance of the curves in Figs. 7 and 8? Are the variations maybe just within the curve's uncertainty range? An error shading including spectral estimation error and diffusion-correction uncertainty would be helpful. From Fig. 7, I would say that visually the 15–20 year band exhibits correlation with the 20–30 year band and to a lesser extent also with the 7–15 year band; can you comment on this? This could either arise from correlation in the spectral uncertainty, or the speculated link to climate is not confined to the 15–20 year band.

As shown in Fig. 7, the diffusion-correction has very little influence on the shape of the curve. We have added shading for the uncertainty on the diffusion correction in Figs. 7 and 8, showing that the error is very small. L188 is also revised to better emphasize this:

"The higher frequency bands are more diffused, and we find that the 3–7 and 7–15 year bands are not reliably corrected after 5.5 ka due to uncertainty in the diffusion length estimate (see Fig. 3). Fig.7 demonstrates that the 15–20 and 20–30 year bands are less influenced by diffusion, as there is a minimal difference between the strength of the raw and diffusion-corrected data, and the diffusion-correction does not significantly influence the shape of the curve over 8 ka."

We would expect that the visual similarities between the 15–20 and 20–30 year bands suggest that the link to climate is not strictly confined to the 15–20 year band, and rather is centered on this band.

Stacks of Greenland ice-core records show stronger variability compared to the background around the 20-year period in general (Chylek et al., 2011), so a comparison with the full RECAP isotope spectrum (either a diffusion-corrected or, if this is tedious, the raw one) would help to better place the new spectral data into context with existing data from the region.

Here we have computed the FFT of the raw (we cannot diffusion-correct the full 8 ka record) RE-CAP $\delta^{18}$O data through 8 ka, smoothed to a 1-year sampling frequency to remove high-frequency noise. We can see that there is a peak at a periodicity of approximately 20 years, similar to that observed in Chylek et al. (2011). This peak may be slightly weaker to that observed in Chylek et al. (2011), as the stack of ice cores used there would increase the signal-to-noise ratio. This figure has been added to the appendix.

[Figure]

Finally, I have some concerns regarding the interpretation of the seasonal isotope data. In section 3.2.3 the authors conclude that the decreasing winter trend in isotope data could be explained by increasing sea ice which correlates with decreasing winter temperatures. In this regard, the authors also cite the paper by Noone and Simmonds (2004). However, as far as I understand this study, the effect of changing sea ice on isotopic composition is rather through controlling the influence of local oceanic moisture in comparison to long-range transport, instead of a direct influence on local temperature. It would thus be worth to discuss the influence of sea ice on isotopic composition in more detail here.

We have reframed section 3.2.3 as a look at regional climate variables, including sea ice. In this case, we separate the effects of sea ice on regional temperature and moisture source. We have replaced citations for Noone and Simmonds, 2004 with more appropriate citations for Greenland ice cores, such as Johnsen et al., 2001; Holme et al., 2019; and Klein and Welker, 2016 where appropriate. The following is the revised section 3.2.3:

"Indirect solar effects may influence other parts of the climate system, which then affect the local climate at Renland. A cooling trend in the North Atlantic is observed over the last 3 ka in records of glacial expansion, ice sheet growth, and increased drift ice (Miller et al., 2010), driven by a decrease in total annual insolation (Kaufman et al. 2009). In the RECAP core, a decrease in $\delta^{18}O$ is observed in winter, while the summer signal remains stable. While it is difficult to determine the cause of this trend without isotope-enabled modeling, we can hypothesize how possible mechanisms involving regional climate variability might influence seasonality of the isotope signal at Renland.

Since Renland is closer to the coast and open ocean than other inland ice cores, sea surface conditions could play a substantial role in Renland climatology (Holme et al., 2019). One possible factor is sea ice extent, which is both influenced by annual insolation (Muller et al. 2012) and influences total absorbed insolation at the surface due to albedo. A number of studies have documented an increase in sea ice cover in the North Atlantic and Fram Strait over the last 3 ka (Mller et al., 2012; Fisher et al., 2006; Jakobsson et al., 2010; Polyak et al., 2010). Sea ice extent has been previously studied through impurities in the RECAP core; iodine concentrations from the RECAP ice core suggest increasing sea ice over the last 3 ka (Corella et al., 2019; Saiz-Lopez et al., 2015), and bromine enrichment has been used to estimate sea ice conditions through 120 ka (Maffezzoli et al., 2019).

The formation of sea ice primarily occurs in winter, and increasing sea ice would be correlated with decreasing regional temperatures in winter and for portions of the shoulder seasons, depending on the timing of ice formation in fall and melt in late spring. A more open ocean regime at 2.6 ka, driven by higher total annual insolation, would keep winters warmer in coastal Greenland due to ocean heat contribution to the atmosphere (Screen and Simmonds, 2010). In recent centuries prior to the Industrial Revolution, lower total annual insolation and increased sea ice would dampen the moderating effect the open ocean has on coastal winter temperatures, resulting in colder winters. Increasing sea ice is therefore consistent with increasingly colder winters at Renland, whereas summers would largely be immune to sea ice response since nearby water bodies have little to no summer sea ice. This may explain the similarity between the winter $\delta^{18}O$ signal and total annual insolation at 71° N (Figs. 10c, 8c).

However, it is nearly certain that other regional climate variables have an influence on the $\delta^{18}O$ signal in the RECAP core. At this time we lack a comparison to seasonality at other locations in Greenland, which would help to determine the extent to which the trends observed in the RECAP record are due to local or regional influences. The RECAP core is unique in that it has both

high sampling resolution (0.5 cm, whereas most other cores are over 2 cm), and high accumulation rate (45 cm yr$^{-1}$ compared to 10–20 cm yr$^{-1}$ inland), allowing for a much more accurate diffusion-correction of the seasonal isotope signal. At Renland, we may also observe the effects of atmospheric and oceanic circulation patterns and sea ice extent, which can control the influence of local oceanic moisture in comparison to long-range transport. This would alter the $\delta^{18}O$ signal through moisture source instead of a direct influence on local temperature (Johnsen et al, 2001; Klein and Welker, 2016). As we do not have records of isotope seasonality from inland Greenland, it is difficult to identify whether an effect such as this uniquely influences the RECAP isotope signal. These factors could be instead be further explored through additional modeling studies. "

Furthermore, the authors also state as a main conclusion that this finding "is a valuable demonstration of the additional regional climate information contained in coastal ice cores" (P18 LL328-329). However, I would be more convinced if you could explicitly show that the RECAP spectral (see above) and seasonal signals are clearly distinct from other Greenland ice cores. I don't see this being convincingly presented. The GRIP seasonal data basically shows the same features (stable summer, decreasing winter; cf. Appendix A) – isn't the fact that the resulting increase in GRIP seasonal amplitude appears "muted" compared to Renland (Fig. A1) simply explained by the difference in axis scaling ($\sim$ 1.5‰ axis range for RECAP compared to $\sim$ 3‰ for GRIP)?

After careful consideration, we have decided to remove the comparison to GRIP from the manuscript. Originally, we thought it was an interesting comparison, but the uncertainty on the GRIP record is too high for any meaningful comparison and it will only introduce confusion when interpreting the RECAP core as a unique coastal ice core record. Here, we outline the reasoning for the uncertainty on the GRIP record, and how we have modified our conclusions on the RECAP record after removing this comparison.

First, the GRIP isotope record has much lower sampling resolution (2.5 cm) than the RECAP record (0.5 cm). This is because the GRIP record was measured using discrete ice samples, prior to the advancement in technology that allowed for the continuous flow method used in analysis of the RECAP core. Additionally, GRIP has a much lower accumulation rate (23 cm yr$^{-1}$ compared to 45 cm yr$^{-1}$ at Renland), resulting in lower resolution sampling with time.

Because GRIP has a much lower sample resolution, this introduces significant uncertainty to the analysis and diffusion correction. The lower accumulation rate also makes the diffusion correction more susceptible to seasonal bias in accumulation. We could rule out this issue at Renland by using the CFM test, but preliminary CFM models of WAIS Divide (which has similar temperature and accumulation rate to GRIP) have indicated a much stronger influence on the isotope signal. For WAIS Divide, additional proxy records such as chemistry data can be used to help constrain any changes in seasonality of accumulation (manuscript by Jones et al., will be submitted soon), but we do not have the needed additional data for GRIP. Therefore, we cannot rule out the possibility that any trends observed at GRIP are not simply due to changes in seasonality of accumulation, making a comparison to RECAP less meaningful.

For example, we have calculated the mean rate of change over the period from 2.6 ka–present for the normalized amplitude. While the rate of change is larger at RECAP (0.075 ka$^{-1}$) than at GRIP (0.034 ka$^{-1}$), the uncertainty on the amplitude for GRIP effectively overwhelms the signal. This is demonstrated by the gray shading in the figure shown here, which is substantially larger for GRIP than for RECAP.

[Figure]

FIGURE 1. Comparison of RECAP and GRIP seasonal data. (a) The summer and winter $\delta^{18}$O signal for both RECAP (red and blue) and GRIP (orange and green). (b) The annual amplitude for RECAP (black) and GRIP (purple) shown as both $\delta^{18}$O (right axis) and normalized to the maximum $\delta^{18}$O for each record (left axis).

Because we do not have an inland ice core with high enough sampling resolution and accumulation to accurately reconstruct the seasonal signal, we cannot claim that the trends observed in the RECAP isotope signal are unique. However, the RECAP record is unique in that it is the only existing Greenland core which we are able to use for this type of analysis, making it a very valuable record. We have revised part of Section 3.2.3 and the Conclusions to reflect this:

Section 3.2.3: "However, it is nearly certain that other regional climate variables have an influence on the $\delta^{18}$O signal in the RECAP core. At this time we lack a comparison to seasonality at other locations in Greenland, which would help to determine the extent to which the trends observed in the RECAP record are due to local or regional influences. The RECAP core is unique in that it has both high sampling resolution (0.5 cm, whereas most other cores are over 2 cm), and high accumulation rate (45 cm yr$^{-1}$ compared to 10–20 cm yr$^{-1}$ inland), allowing for a much more

accurate diffusion-correction of the seasonal isotope signal. At Renland, we may also observe the effects of atmospheric and oceanic circulation patterns and sea ice extent, which can control the influence of local oceanic moisture in comparison to long-range transport. This would alter the $\delta^{18}O$ signal through moisture source instead of a direct influence on local temperature (Johnsen et al, 2001; Klein and Welker, 2016). As we do not have records of isotope seasonality from inland Greenland, it is difficult to identify whether an effect such as this uniquely influences the RECAP isotope signal. These factors could be instead be further explored through additional modeling studies."

Conclusions: "At this time, Renland is the only available Greenland ice core that has high-resolution sampling and high accumulation rates. These factors are necessary to rule out seasonality of accumulation effects on diffusion, allowing for an interpretation of the summer and winter patterns for the last 2.6 ka. Whether Renland is unique in its downward trend in winter values remains to be seen."

**Specific comments**

P1 L7: "The strength of the interannual frequency band decays rapidly". I suggest to remove this result, since it is a direct consequence of firn diffusion and therefore a trivial and expected result for almost any ice core, which is not relevant for the abstract.

Done

P1 LL8-9: "Comparison to other North Atlantic proxy records suggests": As far as I follow the paper, the results concerning the Renland 15–20 yr variability are compared to only one other proxy record, which is the VM28-14 record of Bond et al. (2001), so this statement should be adjusted.

This statement refers not only to VM28-14, but also the discussion including other Greenland ice cores from Holme et al. (2019) and Chylek et al. (2011).

P1 L17: "Greenland ice core records are valuable for determining a more comprehensive picture"; more comprehensive compared to what or which other records? This is not clear from the context provided.

"more" has been removed from this sentence.

P1 L20: "[...] were used in analysis of the Renland ice core". This is a bit confusing since there is also the old Renland ice core which was already drilled in 1988. If I am not mistaken you present and refer here to the new RECAP ice core. Please clarify this throughout the paper.

This is clarified throughout the paper, such that "Renland" refers to the peninsula and its local climate, and "RECAP" refers to the ice core and water isotope record.

P2 Fig. 1: Please introduce the scope of the figure in a first sentence, e.g. "Map of the study region", and add to the caption that the red circle denotes the RECAP (?) drill site, blue circles other Greenlandic ice core sites, and the green circle the location of a marine sediment drift ice proxy record.

The caption has been revised:

"A map of the study region shows the RECAP drill site (red) is located on the east coast of Greenland. The Renland ice cap is approximately 80 km wide, and is isolated from the Greenland ice sheet. The drill site is near the summit of the ice cap, at a location of -26.75, 71.2333. The locations of several other Greenland ice cores are also shown for reference (blue), as well as North Atlantic sediment core VM 28-14 (green) (Pawlowicz, 2020)."

P2 LL28-29: The Noone and Simmonds (2004) paper explicitly investigates the influence of sea-ice cover on western Antarctic ice cores; the statement here would suggest to a reader unfamiliar with the topic that it instead covers the link between coastal Greenland climate and ocean conditions, which is not the case. Please adjust the statement, stating that a similar influence of sea ice as observed in Antarctica could be expected for coastal Greenland cores, or provide a directly relevant reference. This similarly applies to the same reference in Sect. 3.2.3 (P17 L301).

We have replaced citations for Noone and Simmonds 2004 with Holme et al., 2019; Johnsen et al., 2001; and Klein and Welker, 2016, which more appropriately discusses Greenland climate.

P1 L21 – P2 L34: The flow of information is rather incoherent in this paragraph. Please consider rewriting it such that you start to introduce the new RECAP core and then describe the peninsula, the ice cap and the local climate.

This paragraph has been revised:

"The RECAP ice core extends 584 m to bedrock, with the oldest ice dating 120 ka; here we present the Holocene water isotope record (i.e. $\delta^{18}O$) (Fig. 2). Polar ice core water isotope records are correlated to condensation temperature at the time of precipitation (Dansgaard, 1964; Dansgaard et al., 1973; Craig and Gordon, 1965; Merlivat and Jouzel, 1979; Jouzel and Merlivat, 1984; Jouzel et al., 1997), and integrate across regional ocean and atmospheric circulation patterns and sea surface conditions along the moisture transport pathway (Johnsen et al., 2001; Holme et al., 2019).

The Renland Peninsula is located on the eastern coast of Greenland in the Scoresbysund Fjord (Fig. 1). The ice cap is unique in that it is isolated from the Greenland Ice Sheet by steep fjords and is only 80 km wide; as a result, the thickness of the ice cap is constrained and did not experience significant change during most of the Holocene (Vinther et al., 2009; Johnsen et al., 1992). The Renland Peninsula experienced post-glacial uplift in the early Holocene due to ice sheet retreat, but the rate of uplift has been minimal over the last 7 ka (Vinther et al., 2009). These factors imply that the climate record is not influenced by long-term changes in elevation through the mid to late Holocene. The modern accumulation rate at Renland is approximately 45 cm ice equivalent accumulation per year, resulting in clearly defined annual layers. In comparison to inland ice core records, the local climate at the coastal Renland site is more likely linked to sea surface conditions and North Atlantic climatology (Holme et al., 2019; Johnsen et al., 2001)."

P3 LL69-70: "due to a small amount of mixing introduced in the system". Please explain the CFA mixing effect shortly here and either state the amount or provide a reference.

This sentence has been revised:

"Water isotope data has sub-mm nominal resolution, as there is a small amount of mixing introduced in the system due to liquid water mixing in tubing or vapor mixing (Gkinis et al. 2011, Jones et al. 2017)."

P4 L79 – P5 L86: As I understand it, the main reason for the loss in high-frequency variability is thus in both cases diffusion (as expected), but amplified by two distinct reasons. Please restructure the paragraph accordingly to highlight this and mention diffusion as the cause in the first place, and then elaborate the two reasons for the strong diffusion within the LGM and the basal ice.

This paragraph has been revised:

"The RECAP record exhibits a much higher effective resolution in the Holocene than during the glacial period. High-frequency variability is lost rapidly with depth due to an increase in the effect of water isotope diffusion, which occurs for two reasons: 1) At the last glacial maximum, the Laurentide ice sheet extended to approximately 40° N, and the associated temperature decrease and sea ice increase led to significantly drier conditions with minimal precipitation in Greenland. As a result, water isotope diffusion has a much greater effect in thinner glacial ice layers, acting to eliminate high-frequency signals. 2) Extreme basal thinning (see Fig. B1) effectively deforms annual layers into very thin intervals of ice, allowing solid-phase water isotope diffusion to have a disproportionately strong effect on the oldest ice. The glacial period is condensed into approximately 30 m, and no high-frequency climate signals (i.e. annual, interannual, decadal) are preserved. As a result, we focus here on the Holocene record through 10 ka, which has retained greater high-frequency climate information."

P5 LL90-91: "along concentration, temperature, and vapor-pressure gradients"; I would argue that concentration and vapor pressure are directly linked, so stating both is redundant. Just mention concentration, as it is more intuitive.

'vapor pressure' has been removed.

P5 L104: Please provide a reference for the CFA system mixing effect, if not done before (please see respective comment above).

Citations have been added for Gkinis et al. 2011 and Jones et al. 2017 in L70.

P5 LL105-106: "with a 100-year time step"; at first reading, this can be misunderstood. I understand you mean that the step size between the overlapping windows is 100 years, but as it is written it might be confused with the temporal resolution; please clarify. Please also underline that you produce a spectrum for each window. Additionally, it is more common to refer to this quantity as the power spectral density (PSD); please change this throughout the manuscript.

This section has been rewritten, and other instances have been corrected to PSD:

"...overlapping windows corresponding to 300-year time periods with a step size of 100 years between windows. This produces the power spectral density (PSD) (‰$^2$m) vs. frequency ($f$) (m$^{-1}$) for each 300-year window."

P5 L111 – P6 L118: Why is the linear regression in log space only used to assess the uncertainty of the diffusion length but not to estimate the diffusion length value itself? Should this not maybe provide a better fit than a nonlinear model? Or do you assume any other shape for $P_0(f)$ than white noise for the fit? What about the measurement noise – is it subtracted before? You do talk about this some point later in the manuscript, but it should be mentioned here already.

The linear regression in log space and the Gaussian produce the same estimate of diffusion length, but it is easier to tune the fit of the Gaussian to find the most accurate estimate. The measurement noise does not influence the fit of the Gaussian, since only the diffused section (at lower frequencies than the noise) is used for the fit. This is shown in Fig. 3, by the blue line indicating the section used for fitting the Gaussian. LL113-114 has been edited to clarify this:

"Eq. 2 is fit to the diffused portion of the PSD to estimate $\sigma_z$ (Fig. 3a, diffused section indicated by blue line)."

P6 Eq. 3: I am not sure if I understand your uncertainty estimation correctly. From the slope $m$ of the regression of $\ln(P)$ against $f^2$, one obtains the diffusion length as $\sigma = \sqrt{m}/(2\pi)$. If the slope estimate has some uncertainty $\Delta m$, then the uncertainty of the diffusion length should be $\Delta\sigma \sim \frac{\partial\sigma}{\partial m}\Delta m = \frac{1}{2\pi}\frac{1}{2\sqrt{m}}\Delta m$; so the exponent in Eq. (3) seems to be wrong ($-\frac{1}{2}$ instead of $\frac{1}{2}$), and isn't the slope uncertainty missing in there as well?

Eq. 3 is adapted from Jones et al. (2017), which has a more in-depth derivation of the diffusion equations (we decided this would be an unnecessary repetition here, as it is cited). From Jones et al. (2017):
Eq. 3: $\sigma_z = \frac{1}{2\pi\sqrt{2}} \cdot \frac{1}{\sigma_g}$
Eq. 7: $\sigma_g = \sqrt{\frac{1}{2\cdot\mathrm{abs}(m_{lr})}}$
Therefore combining these two equations yields our Eq. 3. The slope uncertainty is noted in L116 as the maximum and minimum slope ($m_{lr}$).

P6 L124-130: Please clarify; it is clear that the diffusion length in depth should decrease due to thinning, but for the sake of clarity it could be worth to mention that, to a first approximation, the firn diffusion length should stay constant in the time domain after pore close-off, so an observed increase in diffusion length in time must have a different cause, i.e. ice diffusion etc.

LL127-128 has been revised:

"...However, the diffusion length in the time domain (Fig. 3d) increases with age. Because the effects of firn diffusion are locked in after the pore close-off depth, an increase in diffusion length with time could be potentially due to effects of solid-phase diffusion. Alternatively, changes in surface conditions..."

P6 L134: There is not really much information regarding the two fitting methods to be found in the appendix or the respective figure caption.

Fig. B2 caption has been revised to include more information about the fitting methods:

"Gaussian fits (green line) can be found for the diffused part of the spectrum (green points) for most 300-year PSD windows, but there are two problematic sections from 5.6–6.7 and 7.8–8 ka. Two fitting schemes are used for these sections to ensure bias is not introduced, both shown here for a window from 6363–6663 yr b2k which exhibits a double-Gaussian shape. Fit 1 (left) utilizes a similar fitting range as surrounding windows to determine the diffusion length. The fitting range for each window is interpolated from the reliable ranges applied to the spectrum on either side of the problematic section. This results in a poor Gaussian fit, but a better estimate of the fitting range. Alternatively, the fitting range for Fit 2 (right) is selected so that the second Gaussian is used to determine the diffusion length. This results in a better Guassian fit, but erratic changes in

the fitting ranges applied to different windows, which is not observed in the reliable spectra."

P6 L139: What about the contribution of non-climatic noise (e.g. from stratigraphic noise; Mnch and Laepple (2018)) to the spectra?

It is possible that there is some stratigraphic noise in the record observed in high frequencies; however, Renland has a much higher accumulation rate than the Antarctic ice core sites studied in Munch and Laepple (2018), and would be expected to have a higher signal-to-noise ratio. This is confirmed in Holme et al (2019), with a comparison of three cores drilled at Renland (1988 core, 1988 shallow core, and RECAP). Therefore we do not think that stratigraphic noise is significantly contributing to our spectral analysis of high frequencies.

P6 L140: I would not call it amplitude at all, since this term refers to truly oscillatory behaviour. Just use "strength" or "strength of the variations". Also it is not clear to me which exact quantity you investigate: do you use the square root of the power at the respective frequencies or averaged over the frequency band? Or do you integrate (for the frequency bands) the power spectrum to obtain the average variance (and then standard deviation from the square root of it) within each band? Please clarify here and also when discussing the results in Sec. 3.1.

We feel it is appropriate to use the term amplitude, as it is referring to specific frequencies and is also defined here as the strength of the signal. For the calculation of relative amplitude/strength of frequency bands, we integrate the power spectrum across the frequencies, and then take the square root to find the amplitude. This is better clarified in L140:

"While the raw isotope data is comprised of a continuum of climate variability at all frequencies, it is possible to separate out the strength (ie. the amplitude) of varying frequency bands within that continuum. The power ($P_i$) of individual frequencies from 1–20 years are identified for each 300-year PSD window, as well as the frequency bands of 3–7, 7–15, 15–20, and 20–30 years. For frequency bands, the average power is calculated by integrating across the power spectrum within each frequency band:

$$(2) \qquad P_i = \frac{\displaystyle\int_{f_a}^{f_b} P(f)\mathrm{d}f}{f_b - f_a}$$

where $f_a$ and $f_b$ represent the lower and upper frequency limits, respectively. Because the frequencies are not evenly spaced, this method ensures that the average is not biased towards higher frequencies. The amplitude (ie. the strength) is calculated as the square root of $P_i$."

P6 LL141-142: "individual frequencies are normalized", "each frequency band is normalized"; better write that not the frequencies are normalized but the PSD/variance at the frequency/within the frequency band.

All instances of this have been corrected.

P7 Fig. 3: Please introduce the scope of the figure in a first sentence, e.g. "Isotope power spectra and diffusion length estimation." Also, can you comment on the rather unusual shape (sharp increase, then flat) of the estimated spectra in panel (a) for large periods (> 3 m); which part of it can be trusted and which might be an artifact of the estimation method? In addition, it might be also worth to mention that the strong increase in PSD of the diffusion-corrected spectrum for large frequencies (small periods; violet curve) arises from blowing up of the measurement/CFA noise by

the diffusion correction.

The flat section of the spectra is likely an artifact, but is at much lower frequencies than we analyze here and so would not influence our results. The x-limits have been adjusted to cut out this portion of the spectrum, since it is not relevant and could introduce confusion. The figure caption has been revised:

"Isotope power spectra and diffusion length estimation. (a) The PSD for a 300-year window from 2263–2563 yr b2k. The diffused section of the spectrum is fit with a Gaussian (Eq. 1), used to estimate diffusion length for each window. The signal is cut off at a frequency of 1.1 yr$^{-1}$ (indicated by vertical black line), after which point it is primarily noise. At periodicities below the cut-off frequency, diffusion correction of noise results in an unstable signal (as shown by the purple PSD to the right of the cut-off frequency). (b) Natural log of the same PSD..."

P8 L154-156 and Fig. 4: I wonder, given the sub-seasonal variability of the raw (diffused) signal, if the smoothness of the deconvolved signal is an artifact of the inversion method and not real? In other words, would we really expect the original (pre-diffusion) seasonal $\delta^{18}$O signal to be so smooth? Have you tried to diffuse again your deconvolved signal and compare it to the original?

The smoothness of the diffusion-corrected signal is a result of cutting off high frequencies, which blow up due to measurement noise. If we were to artificially diffuse the diffusion-corrected signal, we would not be able to add the high frequencies back in and it would result in a smoother signal that the original raw signal.

P8 L164: "and a constant amplitude (4‰)"; why do you choose the amplitude about twice as large as observed from the seasonal isotope data (Fig. 9e)?

The 4‰ sine wave is chosen based on the mean annual signal in the upper 10 years of the core, for which we can assume there is little to no effect from diffusion. Because it is not possible to diffusion-correct the firn column for the last 76 years, this is the closest estimate we can make for the MAR period 1958-1978. LL164-165 has been revised to clarify:

"A constant amplitude (4‰) sine wave is used to represent the annual isotopic variability (Fig. B4c), based on the mean amplitude of the relatively un-diffused most recent 10 years of $\delta^{18}$O signal."

P8 L165: One cannot really see the precipitation bias clearly on Fig. B3; better refer to Fig. 5 here and reference Fig. B3 in the first sentence. Also please consider to list the input information in a more coherent way: Start with the MAR temperature data, then mention precipitation bias in MAR and then your assumed accumulation scenarios.

This section has been rewritten (note that Fig. 5 has been moved to the appendix and combined with Fig. B3; (a–b) refer to what was Fig. B3, and (c–e) refer to what was Fig. 5).

"...provided by the Modèle Atmosphérique Régional (MAR; version 3.9 with monthly ERA forcing) (Fettweis et al. 2017 (Fig. B4a–b) A constant amplitude (4‰) sine wave is used to represent the annual isotopic variability (Fig. B4c), based on the mean amplitude of the relatively un-diffused most recent 10 years of $\delta^{18}$O signal. The mean monthly 1958–1978 temperature cycle (Fig. B4e) from MAR is used for all model scenarios. For the 20-year period, MAR simulates that the summer months July?September receive the most accumulation on average (Fig. B4d). The mean annual

accumulation from the 175 20-year MAR period is used for all model scenarios, with the following variations on the seasonality of accumulation applied:..."

P10 Fig. 6: Please introduce the scope of the figure in a first sentence. Additionally, dissipation in a physical sense refers to the energy loss by friction; please rephrase, e.g. "the annual signal strength decreases"/"the annual signal diffuses".

Fig. 6 caption has been revised:

"The relative amplitude of selected individual frequency bands from 1–20 years is shown for the RECAP $\delta^{18}$O signal, with all bands normalized to the strength of the annual frequency in the most recent window. This demonstrates how rapidly the strength of the annual signal decreases, while lower frequency signals are preserved for a greater period of time."

P10 L185 and Fig. 6 and 7: Why does the normalized amplitude has a unit (‰)?

The ‰ unit has been removed from the y-axis for normalized amplitudes in Figs. 6, 7, 8.

P10 LL186-187: Remove the sentence "Each frequency band is...", since this information is already given in the Methods section and the figure caption.

Done

P12 L219: What kind of "similar 20-year signal"? Between the time series (correlation of e.g. 20-yr averages) or in spectral properties?

This section has been rephrased to clarify the 20-year variability observed in spectral properties:

..."Over the period from 1303–1961, spectral analysis shows 20-year variability in the ice core stack, also observed in prior model simulations of AMO (Chylek et al., 2011; Knight et al., 2005). The correlation between prominent multidecadal variability in both Greenland ice core records and climate model simulations is attributed to..."

P13 L227: "of this relationship"; this is unclear: of which relationship? How does this sentence relate to the statement from the Knight et al. paper directly before?

This is referring to the relationship between the RECAP $\delta^{18}$O 15–20 year variability and AMO; this sentence has been revised to clarify:

"...Additionally, Knight et al. (2005) shows that the AMO signal observed in the HadCM3 model is non-stationary on millennial time scales, potentially explaining the variability in the strength of the 15–20 year signal at Renland. While the correlation between the marine HSG record and the RECAP ice core water isotope record is not conclusive evidence of the relationship between the RECAP 15–20 year $\delta^{18}$O signal and AMO, there is a plausible physical connection..."

P13 L236: Please clarify what you mean with non-stationarity here. You just presented that the winter time series shows a trend, so is non-stationary in the the sense that its mean value is decreasing, but you say that the summer value does not show any clear trend, so it would be stationary. Or do you refer in both cases to other statistical quantities which show time dependence, e.g. the time series variance?

Non-stationary here refers to the time series variance; L236 is revised to remove 'non-stationary' to avoid confusion:

"Both summer and winter records also exhibit some variability on centennial timescales over the last 2.6 ka."

P13 Sect. 3.2.1: This is a well-written section that convincingly argues that accumulation seasonality and changes in melt layers very likely are not the cause for the observed trends in isotope seasonality. The only thing which comes to my mind is, however, the possibility of changes in the seasonality in firn temperature to maybe affect the isotope seasonality via seasonally varying diffusion lengths (Simonsen et al., 2011). Could you provide arguments to constrain this possibility?

Seasonal changes in firn temperature are certainly a valid concern with respect to diffusion length, but we believe that this does not significantly influence our results for a few reasons. First, the CFM does use an annual temperature cycle in the diffusion model. While it is possible that the timing and value of the minimum and maximum annual temperature may vary slightly year-to-year, we would not expect this to impose significant changes in diffusion over a time scale of months. Simonsen (2011) discusses the temperature effect on diffusion with regards to major climate events such as the Bolling-Allerod and deglacation, which would experience much larger changes in mean annual temperature. Additionally, if there is a long-term effect in diffusion length due to changes in mean annual temperature (ie. slightly colder mean temperature at present than at 2.6 ka), this would be reflected in our diffusion length calculations.

P13 LL253-254: How can I see the pre-diffusion amplitude in Fig. 10 to assess the under-correction? Is it identical to the constant accumulation case diffusion-corrected value? Or does this case also lead to an amplitude under-correction?

The pre-diffusion amplitude is identical to the diffusion-corrected constant accumulation case. A statement has been added to L254 has been revised to clarify this:

"Seasonally-biased accumulation scenarios can be compared to the constant accumulation scenario, in which each month receives the same amount of accumulation and the diffusion-corrected amplitude matches the pre-diffusion signal."

P14 Fig. 9: Please introduce the scope of the figure, e.g. "RECAP annual and seasonal $\delta^{18}$O data."

Done.

P15 Fig. 10: Please clarify: "Comparison of diffusion-corrected seasonality...".

Done.

P15 L255: I cannot see an 18% difference in Fig. 10; when I compare the green dots to the black line, there is at most a difference of maybe 7%?

The percent difference is calculated for the annual amplitude, and has been revised to use the constant accumulation scenario as the reference. Previously the difference was calculated as (max amplitude-min amplitude)/(min amplitude) = 18%, and is now calculated as (max amplitude-min amplitude)/(max amplitude) = 15.7%. To make this more clear, two changes have been made.

First, Fig. 10 has been revised to have three panels: summer $\delta^{18}$O, winter $\delta^{18}$O, and annual amplitude $\delta^{18}$O; this way the differences are easier to see. Second, LL255-256 has been revised to clarify that the change is in reference to the constant accumulation scenario annual amplitude:

"Seasonally-biased accumulation scenarios can be compared to the constant accumulation scenario, in which each month receives the same amount of accumulation and the diffusion-corrected amplitude matches the pre-diffusion signal. In comparison to the constant accumulation scenario, there is a maximum 15.7% decrease in the annual amplitude of diffusion-corrected isotope values for varied accumulation scenarios, which is a direct result of the bias in the diffusion-correction."

P16 LL274-275: "and the snow layers are so thick at Renland..."; this is repitition from above, you might consider removing this part.

Done.

P17 LL291-292: It might be worth to elaborate a bit more on this conclusion. Do you mean that the insolation cannot explain the seasonal isotope changes since the summer insolation change is too weak or because there is no winter change, or both? It could also be useful to underline the importance of having seasonal information available due to the accumulation bias in the annual data. How do your results connect to the millennial-scale trends seen over the Holocene in other Greenland ice cores (Vinther et al., 2009)?

Vinther et al. (2009) does not address seasonal isotope changes, making a comparison to our results here difficult. However it is relevant to note that Vinther et al. (2009) shows that no uplift correction is needed for at least the last 7 ka (and minimal prior to that), supporting that our results are not influenced by uplift. This is already noted in LL24-26. LL291-292 has been expanded to clarify the comparison between isotope changes and the insolation model:

"The RECAP seasonal $\delta^{18}$O signal shows a stable summer signal and decreasing winter signal; therefore, expected seasonal temperature trends due solely to changes in insolation are not able to account for the observed changes in seasonal $\delta^{18}$O at the Renland site."

P17 L307: Someone not familiar with the sea-ice edge distribution might wonder whether the area is actually ice free in summer at present?

Revised sentence to clarify:

"...whereas summers would largely be immune to sea ice response since nearby water bodies have little to no summer sea ice."

P17 L308: The first part of this paragraph seems to be a repetition from the previous one; please consider to restructure the two paragraphs.

This section has been extensively restructured; please see previous response to General comments for revised version.

P18 L322 and L324: Given the speculative nature of your conclusions, I would rather nuance this and instead write "...and diffusion correction reveals a decreasing trend..." as well as "We instead suggest that the winter trend is likely...".

Done.

P20 Fig. B2: I must admit that I do not really understand the difference in the fitting approaches from the figure caption or the figure itself. Please explain this in more detail.

Fig. B2 caption has been revised to include more information about the fitting methods:

"Gaussian fits (green line) can be found for the diffused part of the spectrum (green points) for most 300-year PSD windows, but there are two problematic sections from 5.6–6.7 and 7.8–8 ka. Two fitting schemes are used for these sections to ensure bias is not introduced, both shown here for a window from 6363–6663 yr b2k which exhibits a double-Gaussian shape. Fit 1 (left) utilizes a similar fitting range as surrounding windows to determine the diffusion length. The fitting range for each window is interpolated from the reliable ranges applied to the spectrum on either side of the problematic section. This results in a poor Gaussian fit, but a better estimate of the fitting range. Alternatively, the fitting range for Fit 2 (right) is selected so that the second Gaussian is used to determine the diffusion length. This results in a better Guassian fit, but erratic changes in the fitting ranges applied to different windows, which is not observed in the reliable spectra."

**Technical comments**

P1 L1 and title: Please capitalize and hyphenate consistently "East-Central Greenland" throughout the manuscript.

Done.

P1 L4: replace "and the annual..." with "while the annual..."

Done.

P1 L9: Please follow the house standard and use a long (em-) dash for range of numbers, so 15–20 instead of 15-20 (here and throughout the manuscript).

Done.

P1 L16: Please hyphenate phrases such as "Ice-core records" throughout the manuscript; I will mention only some additional instances in the following.

We prefer not to hyphenate "ice-core records". Changes have been made to hyphenate other phrases (see following instances listed).

P1 18-19: Please change to "Cavity Ring-Down Spectroscopy".

Done.

P1 L19: Please hyphenate "high-frequency signals" (throughout the text); see previous comment.

Done.

P1 L21: Please change to "Renland Peninsula".

Done (also changed on L24).

P2 Fig. 1: Please change to "east coast".

Done.

P3 L41: Please change to "northward".

Done.

P3 LL49-50: Please change to "tree-ring data" and "central and western Greenland ice cores".

Done.

P3 LL59-60 and throughout the manuscript: I very much appreciate that you introduce the correct terminology "water isotopologues" here. However, then please be consistent and avoid phrases involving "water isotope"; instead, simply use "isotope" (e.g. isotope signal, isotope record) or use "water isotopologue" or "isotopic composition", where appropriate.

The phrase 'water isotope' is commonly used in ice core literature, and is our preferred phrase for general references to the water isotope record. However we have reduced the number of uses by changing a number of phrases from 'water isotope' to 'isotope' or '$\delta^{18}$O' where appropriate.

P3 L62 and Eq. (1): Change "SMOW" to "VSMOW".

Eq. 1 has been removed, and as such the need for abbreviation VSMOW has been removed. VSMOW is now only referenced to once as 'Vienna Standard Mean Ocean Water'.

P3 L65: Equations are considered to be a part of the sentence, so please start this sentence with lower case "such" (also for Eqs. (2–4)).

Done.

P3 L67: Change cm/min to cm min$^{-1}$.

Done.

P3 L68: For the sake of clarity, please rephrase the sentence to "using two Cavity Ring-Down Spectrometers running in parallel (L2140-i and L2130-i)".

Done.

P4 L74 "0–4035 yr bp": I guess you mean years before present (1950) here; please define this at the first instance and use "yr BP".

We have changed bp to b2k, and defined as 'years before CE 2000'

P4 L79: Please change to "exibits a much higher effective resolution".

Done.

P5 L97 "804.3 kg/m$^3$": there is no need to use such a precise number here (since it is an estimated value); please write "$\sim 804$ kg m$^{-3}$".

Done.

P5 L102-103: I think one most relevant reference to introduce the diffusion length would be sufficient (e.g. the first paper which introduced the concept in the context of firn diffusion).

Extra references are removed, keeping only Johnsen et al., 2000.

P5 L105: Change to "in the depth domain".

Done.

P5 L106: Change "cycles/m" to "m$^{-1}$".

Done.

P5 L108: Change to "To estimate the diffusion length" or "To estimate diffusion lengths".

Done.

P5 Eq. (2) and P6 L115: Use upright font for mathematical functions such as "exp" and "ln".

Done.

P6 L115: Change to "of the data".

The clarification "of the diffused section of data" is necessary as the linear regression is fit to only a section of the spectrum, as indicated in Fig. 3b.

P6 L124: Change to "m yr$^{-1}$".

Done.

P8 LL159-160: For the sake of clarity, please insert "to the resulting summer and winter time series" after "is applied".

Done.

P9 L177: Change to "diffusion-correct".

Done.

P10 L181: For the sake of clarity, please insert "with time" after "decays rapidly".

Done.

P10 L182: Change to "diffusion-corrected".

Done.

P10 L186: For what is the Jones et al. reference needed here?

This method of analysis was used in Jones et al., 2018; the reference has been moved to L140 in the methods section.

P11 Fig. 7 and P12 Fig. 8: The lines of the two fits (solid purple and solid dashed) are indistinguishable; please use another color for highlighting.

The purple dashed line has been changed to light purple dashed, and it has been clarified in the caption that the lines mostly overlap as the two fitting scenarios result in nearly identical diffusion-corrected signals.

P13 L243: Change to "diffusion-correct".

Done (changed all instances of 'diffusion-correct').

P13 L249 and P15 L256: Change to "diffusion-corrected".

Done (changed all instances of 'diffusion-corrected').

P15 L259 "in which we are aware": I am not familiar with this phrase; do you mean "which we are aware of"?

Changed to "of which we are aware".

P16 Fig. 11 caption: Please clarify: "Changes in Holocene insolation..."

Changed to "Changes in late Holocene insolation...".

P16 L279: Change "of top of atmosphere" to "of the top-of-the-atmosphere".

Done.

P17 L305: Change Fall->fall and Spring->spring.

Done.

P17 L316 "The Renland ice core": Please clarify "The RECAP ice core".

Done.

P19 Figure A1 caption: Please rephrase: "Comparison of Renland and GRIP seasonal data. GRIP experiences...".

Done.

P20 Figure B1 caption: Please clarify: "Mean annual layer thickness in the RECAP core...".

Done.

P20 Figure B2 caption, second line: "to determine a diffusion": Do you mean "to determine the diffusion length"?

Changed to "to determine the diffusion length".